# WEBSHAPER: AGENTICALLY DATA SYNTHESIZING VIA INFORMATION-SEEKING FORMALIZATION

**Zhengwei Tao**[1,2,4,5,*]   **Jialong Wu**[1,2,3,4,*]   **Wenbiao Yin**[4]   **Pu Wu**[1]   **Junkai Zhang**[4]
**Baixuan Li**[3,4]   **Haiyang Shen**[1,4]   **Kuan Li**[4]   **Liwen Zhang**[4]   **Xinyu Wang**[4]
**Yong Jiang**[4,†]   **Pengjun Xie**[4]   **Fei Huang**[4]   **Jingren Zhou**[4]   **Wentao Zhang**[1,2,5,†]

[1]Peking University   [2]Beijing Key Laboratory of Data Intelligence and Security (Peking University)
[3]Southeast University   [4]Tongyi Lab, Alibaba Group   [5]Zhongguancun Academy
tttzw@stu.pku.edu.cn   wentao.zhang@pku.edu.cn

[*]Equal Contribution   [†]Corresponding Author

## ABSTRACT

The advent of Large Language Model (LLM)-powered agents has revolutionized artificial intelligence by enabling solutions to complex, open-ended tasks through web-based information-seeking (IS) capabilities. The scarcity of high-quality training data has limited the development of IS agents. Existing data synthesis approaches typically adopt an *information-driven* paradigm that first collects information and then refines question-answer pairs through retrieval. However, this may lead to inconsistency between information structure and reasoning structure, as well as between the question and the corresponding answer. To mitigate, we propose a *formalization-driven* IS data synthesis framework WebShaper, which systematically formalizes IS tasks using set-theoretic constructs. Central to the formalization is the concept of Knowledge Projections (KP), which enables precise control over reasoning structure by KP operation compositions. During synthesis, we begin by creating seed tasks, then use a multi-step expansion process. At each step, an agentic Expander expands the current formal question more complex through retrieval and validation tools grounded in our formalization. We train our model on the synthesized dataset. Experiment results demonstrate that WebShaper achieves state-of-the-art performance among open-sourced IS agents on competitive benchmarks.

## 1 INTRODUCTION

The emergence of Large Language Model (LLM)-powered language agents has marked a paradigm-shifting advance in artificial intelligence, enabling transformative solutions to previously intractable challenges across domains (Guo et al., 2024; Wang et al., 2024; AutoGPT, 2023; Wu et al., 2023; Ye et al., 2023). Information-seeking (IS) represents a core component of the cognitive autonomy of language agents. This capability not only underpins their adaptability in open-ended tasks but also powers a range of powerful commercial systems such as Deep Research of OpenAI (OpenAI, 2025), Gemini (Gemini, 2025), and Perplexity (Perplexity, 2025).

Current agentic systems for unlocking this capability typically follow a well-established pipeline in agent development: (1) First, construct task-specific trajectories of question-answer pairs; (2) Employ supervised fine-tuning (SFT) to acquire foundational skills (Sun et al., 2025). (3) Generalize strategic decision-making through on-policy reinforcement learning (RL) (Jin et al., 2025). The entire development of the IS agent originates from and its ultimate effectiveness depends on high-quality IS task training data. However, due to its complexity, such a high-quality dataset is both sparse and difficult to construct through crowdsourcing. **Thus, constructing training data through a carefully designed agent pipeline becomes the cornerstone of effective IS agent development.**

Existing IS dataset synthesis methods typically involve freely pre-searching for information online and employing LLMs to generate questions from the collected content. These approaches first organize the collected information into structured formats, then prompt the LLM with the structured data to produce natural language (NL) questions. Their core objective is to map *information structures* into *reasoning structures* within the resulting NL questions. Representative methods like WebDancer (Wu

et al., 2025a) and TaskCraft (Shi et al., 2025) generate linear information chains, while others construct graphs connected via web links (Wu et al., 2025b) or entity coreference networks (Li et al., 2025a). However, these information-driven approaches face two critical limitations. **First**, the synthesis using LLM may struggle to fully comprehend the information structure, resulting in inconsistent reasoning structures or incorrect answers to the generated NL questions. **Besides**, disordered information retrieval will lead to excessive data processing and will collect redundant homogeneous information structures, which limits the diversity of information structures.

To overcome these limitations, we propose WebShaper[1], a formalization-driven IS data synthesis paradigm. Unlike prior approaches, WebShaper first formalize information-seeking tasks and then systematically guide data synthesis through this formalization. During generation, information collection is explicitly controlled by formal task requirements. This framework offers three key advantages: ***Broader Task Coverage***: Systematic exploration of task formalizations enables synthesizing diverse information-seeking patterns unconstrained by pre-retrieval content limitations; ***Task Controllability***: Explicit formalization parameters allow precise specification of reasoning structures and complexity levels; ***Structural and Answer Consistency***: Due to the inherent interpretability and verifiability of formalized representations, synthesized outputs exhibit fewer inconsistencies across both information-reasoning structures and question-answer pairs. With this structured guidance, we produce consistent reasoning and redundancy while ensuring rich, diverse reasoning logic.

At the core of our framework lies a formalization of IS tasks, which enables principled and systematic generation of task instances with controllable collection complexity and reasoning structures. Unlike relevant fields, where there exists task formalization in advance, such as Lean 4 language (Moura & Ullrich, 2021) in math proving and propositional logic in knowledge-centric question answering (Xia et al., 2025), there's no established formalization for information-seeking. We treat IS as a unified problem space where task is systematically derived from compositions of basic units termed Knowledge Projections (KP). To align with the formalized structure, we initiate synthesis by constructing foundational seed tasks, followed by a multi-step expansion grounded in our formal framework. This process employs a dedicated agentic Expander module designed to interpret task requirements via KP representations. At each expansion stage, the expander transforms the current formal question into a more complicated one. It implements layer-wise expansion mechanisms that minimize redundancy while preventing reasoning shortcuts through controlled complexity progression. This process ensures a broad coverage of the formalized task space and the correctness of the question and answer.

We conduct extensive experiments to validate WebShaper dataset by training agents. Comparison with the existing training dataset shows the effectiveness of WebShaper. WebShaper achieves best performances among all open-source IS agents on the GAIA and WebWalkerQA benchmarks. Further discussions demonstrate the validity of each module of our method. We summarize contributions:

- We introduce WebShaper, a formalization-driven data synthesis method for information-seeking agents, grounded in our proposed task formalization. Leveraging this method, we construct the WebShaper dataset, which enables systematic generation of IS instances.
- We propose an agentic Expander that iteratively generates and validates questions in alignment with the formalization, yielding broad coverage IS training data.
- We conduct extensive experiments across multiple benchmarks to evaluate WebShaper. Empirical results demonstrate that models trained with WebShaper consistently outperform baselines, confirming the value of our formalization and synthesis approach.

## 2 INFORMATION-SEEKING FORMALIZATION

In this section, we introduce our formalization of the information-seeking task. We illustrate an example in Figure 1. An information-seeking task $q(T)$ aims to search for knowledge and facts prompted by given facts and locate the answer entity set $T$. For a basic example also shown in Figure 1: $q(T) = $ *Which player of a team in the 2004-05 season, who was born in 90s? This team is founded in 1966 and is an East German football team.* To solve it, one should seek information about *This team is founded in 1966 and is an East German football team* to find that the team is *Berliner FC Dynamo*. And then seek for players of *Berliner FC Dynamo team* in 2004 and 2005 respectively and *players born in 90s*, then reason the answer $T = \{Robert\ Rudwaleit,\ Danny\ Kukulies,\ ..\}$.

---

[1]Without loss of generality, we use WebShaper to denote our data method, dataset, and model.

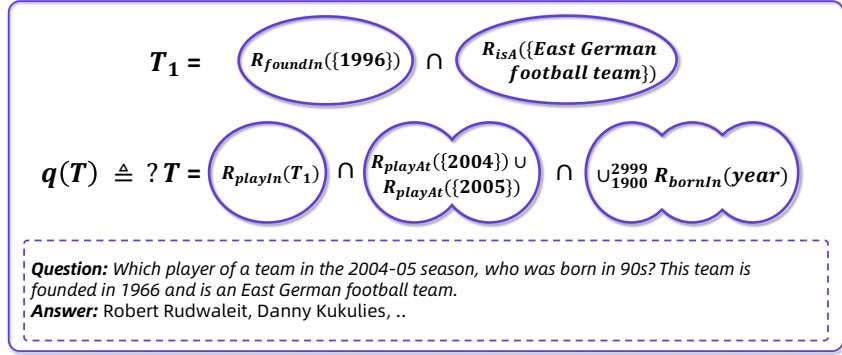

Figure 1: A question-answer case in our information-seeking formalization. We use the **purple** diagram to represent a knowledge projection, which is a set of entities.

Let $\mathcal{E}$ denote the universal set of entities (e.g., players, teams, years). Let $R \subseteq \mathcal{E} \times \mathcal{E}$ denote a subspace of entity pairs where they have a certain relation. For example, if the relation is *bornIn*, $R$ stands for all pairs of (*person*, *year*) where *person* is born in *year*.

For a subset $V \subseteq \mathcal{E}$ and a sub-space $R$, define a Knowledge Projection (KP):
$$R(V) = \{u \mid \exists v \in V, \ (u,v) \in R \text{ or } (v,u) \in R\}. \tag{1}$$
For example, when $R$ denotes entity pairs of relation *bornIn*, $R(\{90s\})$ represents the set of all people born in 90s. **A KP is the set of entities under a certain relation to other entities, which is the basic unit in an information-seeking task.** KP has two operations:

$R$-**Union** $\cup$    In IS, the question may be seeking for a broader condition due to uncertainty about the target. For instance, we only know the target player was playing between 2000-2010 rather than the exact year in advance. The condition can not be more specific than a year range.

Therefore, given $S_1, S_2$ be entity sets and $R$, then:
$$R(V) = R(S_1) \cup R(S_2) \cup \cdots \cup R(S_m) \tag{2}$$
represents $R(V)$ is the union result set in which the entities have a certain relation to entries in either $S_1, S_2, ..., S_m$. If $R$ stands for relation *playAt*, then the set of players who play between 2000-2010 is $R(\{2000\}) \cup R(\{2001\}) \cup \cdots \cup R(\{2010\})$.

**Intersection** $\cap$    Some IS tasks require the target to satisfy several conditions simultaneously. It's interpreted as an Intersection operation of KP:

$$R(V) = R_1(S_1) \cap R_2(S_2) \cap \cdots \cap R_n(S_n) \tag{3}$$
where $R_i$ are about different relations. For example, if $R_1$ is about *playAt* and $R_2$ is about *bornIn*, then $R_1(\{2000\}) \cap R_2(\{90s\})$ stands for players playing in *2000* and born in *90s*.

Based on $R$-Union and Intersection operations, we define $T$ as a target set:

$$T = \bigcap_{i=1}^{p} (R_i(S_{i,1}) \cup R_i(S_{i,2}) \cup \ldots R_i(S_{i,t_i}))). \tag{4}$$

$S_{i,j} \subset \mathcal{E}$. More generally, $T$ can be recursivelly derived by replacing $S_{i,j}$ with other target set as:
$$T = R_1(T_1) \cap R_2(T_2) \cap \ldots \cap R_k(T_k) \tag{5}$$

An IS task $q(T) \triangleq ?T$ is to find what entities a questioned $T$ contains Therefore, the question example can be formalized as shown in Figure 1.

## 3    DATA SYNTHESIS

In this section, we describe the process of our data synthesis with our task formalization. As Eq. (4) shows, an IS task is recursively composited by knowledge projections. In order to better fit the IS task formalization, we start with constructing a seed task, followed by a multi-step expansion approach. This expansion process is built upon our formalization. We then introduce an agentic Expander. It can understand the task formalization with our KP representation. At each expansion step, we implement the layer-wise expansion to reduce redundancy and reasoning shortcuts. The Expander

autonomously retrieves knowledge from the internet, constructs and validates the new FPs to obtain the new question. We elaborate on this process in the following sections.

## 3.1 SEED QUESTION CONSTRUCTION

The first stage is acquiring a substantial volume of diverse and non-trivial seed questions. To enhance acquisition efficiency, we constructed an offline Wikipedia database by downloading all URLs corresponding to Wikipedia articles while preserving the hyperlinks between them. Subsequently, we perform random walks across these articles through their preserved connections. By aggregating the content from articles traversed during these random walks, we utilize an LLM to generate synthetic data instances. Critically, the generated question-answer pairs must be entirely grounded in the content from the collected articles, without relying on external knowledge sources.

However, the resulting seed questions could be noisy and contain hallucinations. We launch a filtering process. We complete all the seed questions by WebDancer framework (Wu et al., 2025a) based on the QwQ model (Team, 2025). We perform 5 times rollouts for each question and keep the data where there must be at least one rollout correctly answering the question. We finally construct 18k seed questions. We denote the harvested seed question as $q^1(T)$.

## 3.2 AGENTIC EXPANSION

Subsequently, we progressively expand seed questions into increasingly complex ones through $l$-step expansion $q^{l+1}(T) = \text{Expand}(q^l(T))$ guided by the task formalization. However, the IS formalization in Eq. (4) is complicated. The nature of recursion and the composition of multiple operations are hard to understand. Besides, since the synthesis relies on retrieving new knowledge online, there are several intermediate processes, such as knowledge filtering and selection. Therefore, we establish an Agentic Expansion. The core of the expansion is the Expander, which is an agent itself to autonomously retrieve information and validate the generation. We introduce the KP representation for the Expander to understand our IS formalization. Then, we propose the Layer-wise Expansion Strategy to mitigate the limitations of redundant and reasoning shortcuts.

### 3.2.1 KP REPRESENTATION

Since $q(T)$ contains recursion and composition of $R$-Union and Intersection operations, it's not trivial to represent $q(T)$ in the Expander agent prompt. We introduce our KP Representation. The key to this representation is to: 1) represent a KP unit. 2) can handle $R$-Union and Intersection operations. 3) can handle recursions of KPs. We introduce Constant and Variable. A constant is a subset of $\mathcal{E}$ explicitly defined by its elements, e.g., $\{90s\}$, $\{2004, 2005\}$. A variable is a subset of $\mathcal{E}$ whose elements are not explicitly given. It may appear as a symbolic placeholder in an expression.

We use a triplet $[X, r, S]$ to represent a KP $R(S)$. $r$ is the name of the relation $R$. $X$ is a variable while $S$ can be a variable or a constant. We use the prefix $V@$ followed by a variable to denote the variable $V$. We use the prefix $@C$ before its natural language description to represent a constant. For example, $R_{bornIn}(\{90s\})$ is represented as $[@V, bornIn, 90s]$. The Intersection operation in Eq.(3) can be naturally represented as a list of triplets $[[X, r_1, S_1], [X, r_2, S_2], ..., [X, r_n, S_n]]$.

For the $R$-Union in Eq.(2), simply expressing it in a list-like form will make the representation complicated in recursive $R$-Union and Intersection. We notice $R$-Union has the following proposition:

**Proposition 1.** *For a certain $R$, $R$-union satisfies the distributive Law:*

$$R(S_1) \cup R(S_2) = R(S_1 \cup S_2) \tag{6}$$

We leave the proof in the Appendix C. With this proposition, we represent the $R$-Union of KP by a merge set $S_1 \cup S_2$. In practice, we express the union of sets by induction (eg. $\{1990\} \cup \{1991\} \cup, \ldots, \cup \{1999\}$ as $\{90s\}$). Or simply add underlines between them (eg. $\{1990\} \cup \{1991\}$) as $\{1990\_1991\}$). After that, our representation would only have an intersection between triplets.

By introducing variables, our representation naturally handles KP recursion by faltten it into the intersection of KPs. For example, given a recursion $R^1(R^2(S))$, we can represent it as $[[V@X, r_1, V@Y], [V@Y, r_2, S]]$. Finally, an IS task $q(T)$ can be represented by a list of triplets. The example question in Figure 1 can be represented as:

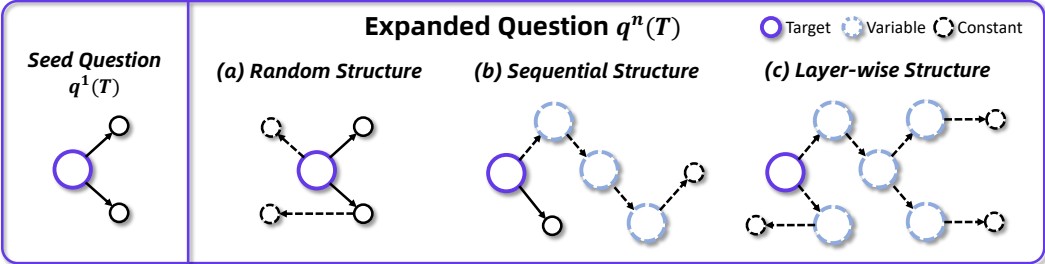

Figure 2: **(a) Random Structure** denotes expanding by randomly adding constants. **(b) Sequential Structure** is expanding on a chain of reasoning sequence. **(c) Layer-wise Structure** traverses layer-wisely on leaf constants and replaces them with variables. "Target" stands for target variable.

$$q(T) \triangleq ?T \quad s.t. \quad [[V@\text{T}, \text{playIn}, V@X], \quad [V@\text{T}, \text{playAt}, C@2004\_05],$$
$$[V@\text{T}, \text{bornIn}, C@90s], \quad [V@X, \text{foundIn}, C@1966], \quad (7)$$
$$[V@X, \text{isA}, C@\text{East German football team}]]$$

### 3.2.2 LAYER-WISE EXPANSION STRATEGY

After representing $q(T)$, we outline the iteration-based expansion process. Compared to prior methods extending questions in natural language, our IS task formalism enables systematic structural analysis, revealing latent patterns and enabling controlled, optimized expansion. To clearly illustrate the expansion strategy, we show our KP representation in a graph: nodes are variables/constants from triplets; edges are relations. For example, Eq. (7) maps to the seed question in Figure 2, where the target variable is determined via given constants. Earlier approaches lacked formal structure, producing either **Random** (Wu et al., 2025b; Shi et al., 2025) (adding FP to arbitrary nodes, Figure 2(a) or **Sequential** (Wu et al., 2025a) (linear reasoning chains, Figure 2(b), both with issues:

- *Redundancy*: Constants directly linked to other constants add sentences like "Dynamo Berlin is a football club based in Berlin" without extending reasoning.
- *Reasoning Shortcut*: Constants that are close to or directly connected to the target result in skipping deeper reasoning for the agent.

We address this via Layer-wise Expansion as illustrated in the Figure 2(c). We layer-wisely traverse the graph to find all leaf constants. When we obtain all the leaf constants of the current graph, an Expander takes each constant once to construct this constant into new FPs. These FPs can form a sub-question that regards the constant as the answer. The expander then merges the sub-questions with the current one to form a new one. It replaces those constants with the sub-questions. Note that the answer for the expanded question always remains. The resulting structure would not have the Redundant and Reasoning Shortcut problems. The number of expanding layers $l$ is a hyperparameter for controlling the task coverage and difficulty.

### 3.2.3 EXPANDER AGENT

We now introduce the Expander, an autonomous agent designed to enhance question generation through iterative refinement. Given an input constant, the Expander first retrieves relevant information, then formulates a semantically coherent sub-question. This sub-question is subsequently integrated with the original query to construct an enriched, context-aware question that better aligns with the underlying information-seeking objective. It builds on ReAct (Yao et al., 2023), cycling through `Thought`–`Action`–`Observation` triples $(\tau_t, \alpha_t, o_t)$, where each `Action` $(\tau, \phi)$ specifies a tool and parameters. We equip the Expander with the following tools:

- `Search`: It enables Expander to conduct Google search by severl queries about a constant and obtains search results. The parameters of this tool are $\phi = \{queries, filter\_year\}$, enabling temporal filtering of search results. This tool returns top relevant URLs and their snippets.
- `Summarize`: This is the key to $R$-Union oepration. This action allows Expander to visit multiple URLs for the constant and summarize the content. The summarization would integrate the retrieved information to obtain a union constant set as stated in Eq.(6). The parameters of this tool are $\phi = \{urls, goal\}$. This tool returns the summarization from the given URLs.
- `Validate`: When Expander completes retrieving and summarizing the KPs of constant, it derives a sub-question and uses this tool to validate the results based on our formalization. The validation

Table 1: **Main results** on GAIA and WebWalkerQA benchmarks. We compare WebShaper with several cutting-edge baselines methods. **bolded** number stands for the best results on the corresponding settings. Blue scores are the highest among all open-sourced methods.

| Backbone | Framework | GAIA | | | | WebWalkerQA | | | |
|---|---|---|---|---|---|---|---|---|---|
| | | Level 1 | Level 2 | Level 3 | Avg. | Easy | Medium | Hard | Avg. |
| *No Agency* | | | | | | | | | |
| Qwen-2.5-7B | Base | 12.8 | 3.8 | 0.0 | 6.8 | 1.25 | 0.8 | 0.7 | 0.8 |
| Qwen-2.5-32B | Base | 20.5 | 9.6 | 8.3 | 13.6 | 3.8 | 2.5 | 3.3 | 3.1 |
| | RAG | 12.8 | 11.8 | 8.3 | 11.8 | 23.1 | 14.3 | 11.3 | 15.3 |
| Qwen-2.5-72B | Base | 20.5 | 13.5 | 0.0 | 14.6 | 9.4 | 7.1 | 3.3 | 6.3 |
| GPT-4o | Base | 23.1 | 15.4 | 8.3 | 17.5 | 6.7 | 6.0 | 4.2 | 5.5 |
| QwQ-32B | Base | 30.8 | 15.4 | 25.0 | 22.3 | 7.5 | 2.1 | 4.6 | 4.3 |
| | RAG | 33.3 | 36.5 | 8.3 | 32.0 | 36.9 | 26.1 | 33.5 | 31.2 |
| DeepSeek-R1-671B | Base | 43.6 | 26.9 | 8.3 | 31.1 | 5.0 | 11.8 | 11.3 | 10.0 |
| *Close-Sourced Agentic Frameworks* | | | | | | | | | |
| | *OpenAI DR* | 74.3 | 69.1 | 47.6 | 67.4 | - | - | - | - |
| *Open-sourced Agentic Frameworks* | | | | | | | | | |
| Qwen-2.5-32B | Search-o1 | 33.3 | 25.0 | 0.0 | 28.2 | - | - | - | - |
| | WebDancer | 46.1 | 44.2 | 8.3 | 40.7 | 44.3 | 46.7 | 29.2 | 38.4 |
| | **WebShaper** | 61.5 | 53.8 | 16.6 | **52.4** | 58.1 | 51.4 | 47.0 | **51.4** |
| QwQ-32B | Search-o1 | 53.8 | 34.6 | 16.6 | 39.8 | 43.1 | 35.0 | 27.1 | 34.1 |
| | WebThinker-Base | 53.8 | 44.2 | 16.6 | 44.7 | 47.2 | 41.1 | 39.2 | 41.9 |
| | WebThinker-RL | 56.4 | 50.0 | 16.6 | 48.5 | 58.8 | 44.6 | 40.4 | 46.5 |
| | Simple DS | - | - | - | 50.5 | - | - | - | - |
| | WebDancer | 61.5 | 50.0 | 25.0 | 51.5 | 52.5 | 59.6 | 35.4 | 47.9 |
| | **WebShaper** | 69.2 | 50.0 | 16.6 | **53.3** | 55.8 | 49.2 | 45.4 | **49.7** |
| Qwen-2.5-72B | WebSailor | - | - | - | 55.4 | - | - | - | - |
| | **WebShaper** | 69.2 | 63.4 | 16.6 | **60.1** | 56.2 | 52.1 | 49.5 | **52.2** |

purposes are to determine: 1) whether the derived sub-question are approximately consistent with the constants based on the formalization. 2) whether it is too simple that can be directly answered by an LLM. This tool would return detailed validation results as `Observation`, and the Expander would take the next action according to it.

## 3.3 TRAJECTORY CONSTRUCTION

We then construct task-solving trajectories using `ReAct` format. At each step, the agent generates `Thought`, performs `Action`, receives `Observation`, and chooses the next move. At each time step $t$, the agent execution loop can be formalized as a triple $(\tau_t, \alpha_t, o_t)$, where $\tau_t$ denotes the free-form `Thought`, $\alpha_t$ represents the structured `Action`, and $o_t$ corresponds to the `Observation` returned by the environment. The Thought component $\tau_t$ is unrestricted natural-language reasoning that the model uses for planning, decomposition, self-reflection, or grounding intermediate assumptions. The Action $\alpha_t$ is further decomposed into an action type $\alpha^m$ and its parameter set $\alpha^p$, i.e., $\alpha = (\alpha^m, \alpha^p)$. The action type $\alpha^m \in \{Search, Visit, Answer\}$ corresponds to the core tool interfaces used in deep information-seeking tasks[2].

To standardize trajectories and facilitate supervised learning, we adopt explicit structural markers for each segment. `Thought` segments are enclosed by <think> and </think>, `Action` segments by <tool_call> and </tool_call>, and `Observation` segments by <tool_response> and </tool_response>. The final `Action` segment, corresponding to the model's ultimate response to the task, is encapsulated in <answer> and </answer>. These markers make agent behavior transparent and machine-parsable, enabling precise control, analysis, and dataset construction. Each question gets 5 rollouts. We remove the trajectories where the answers are wrong, contain hallucinated observations, or severe repetition. We finally obtain 5,000 trajectories for supervised and reinforcement learning.

---

[2]The details of tools are shown in App. G

### 3.4 AGENT TRAINING

To train our information-seeking agent, similar to WebDancer (Wu et al., 2025a), we implement supervised fine-tuning (SFT) followed by reinforcement learning (RL). In SFT, we mask out loss from observation leading to loss. For RL algorithm, we use GRPO (Shao et al., 2024). We leave the details in the Appendix F.

## 4 EXPERIMENTS

### 4.1 EXPERIMENTAL SETUPS

We evaluate WebShaper on two information-seeking benchmarks: **GAIA** (Mialon et al., 2023) and **WebWalkerQA** (Wu et al., 2025b). We use the *LLM-as-Judges* paradigm to evaluate both tasks using the `Pass@1` metric, following Li et al. (2025c). We elaborate on the WebShaper data statistics in the Appendix D. To assess dataset quality, we conduct a comparative study during the SFT stage by training on different data sources. The detailed descriptions of the datasets used in this comparison are provided in Appendix E.

### 4.2 BASELINES

We mainly compare our method to open-source cutting-edge deep research agent frameworks: Search-o1 (Li et al., 2025b), WebDancer (Wu et al., 2025a), WebThinker (Li et al., 2025c), SimpleDeepResearch (Sun et al., 2025), and WebSailor (Li et al., 2025a). As more strict comparing settings, we train baseline models on WebWalkerQA (Wu et al., 2025b), E2HQA (Wu et al., 2025a), and MHQA (Sun et al., 2025), respectively.

### 4.3 MAIN RESULTS

We compare WebShaper with cutting-edge baselines. The results are shown in Table 1. WebShaper achieves best performances on open-sourced methods on both GAIA and WebWalkerQA. Among all GAIA results, WebShaper-on Qwen-2.5-72B excels second-best method WebSailor 4.7 score. On WebWalkerQA WebShaper obtains the highest 52.2 score. WebShaper performs the best on each backbone setting. These results indicate the generalizability of the synthesized data on different models. WebShaper is currently the only open source method with a score of more than 60 points, which is close to the SOTA OpenAI DR system. WebShaper is implemented fully under open-sourced LLMs, demonstrating that high-quality IS data can deeply stimulate the ability of DR Agents. Notably, we find reinforcement learning on QwQ-32B is not significant. Therefore, we report the SFT result on QwQ-32B in Table 1.

### 4.4 DISCUSSIONS

#### 4.4.1 DATA COMPARISON

Table 2: **SFT Data Comparison** on GAIA benchmarks. The best results are in **bolded**.

| Backbone | Dataset | Avg. |
|---|---|---|
| Qwen-2.5-32B | WebWalkerQA | 32.0 |
| | E2HQA | 39.8 |
| | MHQA | 35.9 |
| | **WebShaper** | **43.6** |
| Qwen-2.5-72B | WebWalkerQA | 38.8 |
| | E2HQA | 44.6 |
| | MHQA | 43.6 |
| | **WebShaper** | **45.6** |
| QwQ-32B | WebWalkerQA | 45.6 |
| | E2HQA | 45.6 |
| | MHQA | 41.7 |
| | **WebShaper** | **53.3** |

In this section, we compare WebShaper with baseline datasets. We sample 5,000 data points from each dataset. Then we supervised fine-tune Qwen2.5-32B, Qwen2.5-72B (Yang et al., 2024), and QwQ (Team, 2025) on each dataset. The comparative results on GAIA presented in Table 2 demonstrate the superior performance of WebShaper across all backbone architectures on the GAIA benchmarks. Notably, WebShaper achieves the highest average scores for Qwen-2.5-32B, Qwen-2.5-72B, and QwQ-32B, respectively, significantly outperforming baseline datasets like WebWalkerQA and MHQA. Even when comparing models with similar parameter counts (e.g., Qwen-2.5-32B), WebShaper-enabled models show substantial improvements. The consistency of WebShaper's performance improvement suggests its effectiveness in enhancing model capabilities regardless of architectural design. These findings validate

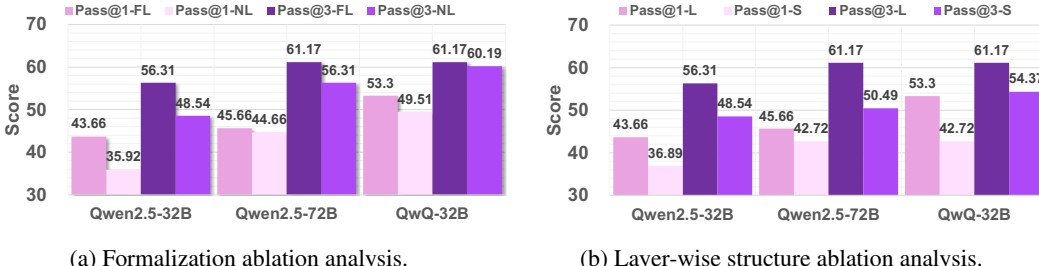

(a) Formalization ablation analysis.     (b) Layer-wise structure ablation analysis.

Figure 3: Discussions on formalization and layer-wise structure.

the effectiveness of formalization-driven data synthesis, making it a superior training data solution for information-seeking tasks. More details are in the Appendix I.

### 4.4.2 RL STIMULATION

We compare GAIA performances between models trained after SFT and reinforcement learning. RL models are trained based on the SFT results. As illustrated in Figure 4, our experimental results demonstrate significant performance improvements across both Qwen2.5-32B and Qwen2.5-72B models after RL training on both GAIA and WebWalkerQA. The Pass@1 metric shows notable enhancements of +7.8 points for the 32B model and an even more pronounced +13.5 points increase for the 72B variant on GAIA. On WebWalkerQA, WebShaper also improves IS capability on a large scale. This substantial gain highlights the critical role of RL in activating advanced information-seeking capabilities within LLM. The breadth and complexity of tasks introduced by our task formalization stimulate dynamic IS strategies during RL. Unlike generic datasets, our carefully curated scenarios require the model to iteratively query relevant information, effectively "training" it to prioritize contextually aligned knowledge fragments.

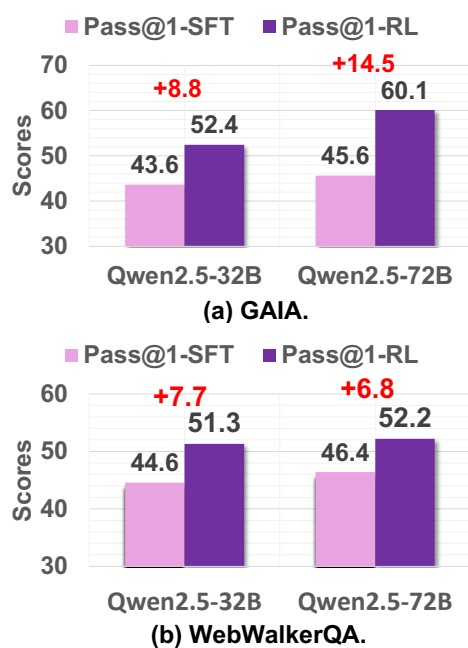

Figure 4: Comparison with SFT and RL.

### 4.4.3 FORMALIZATION

In this part, we validate whether our formalization truly improves the dataset. We compare our dataset to a variation that uses natural language during the data synthesis. We denote our method with formal language as FL, while natural language as NL. This variation takes the current question in each iteration and also uses the Expander agent to expand it to a new question. The Expander process in natural language as well. We SFT Qwen2.5-32B, Qwen2.5-72B, and QwQ on both datasets. The other training setting remains the same. We compare the training results with the variation as shown in Figure 3a. FL excels NL in all base model backbones. These results indicate that our formalization language can mitigate the limitations incurred by natural language. Our IS task formalization can synthesize more forms of tasks. It also reduces error propagation in the synthesis process, leading to consistent and precise question-and-answer pairs.

### 4.4.4 LAYER-WISE EXPANSION STRATEGY

We evaluate the effectiveness of the Layer-wise structure. In order to compare, we set up a variation which uses the same Expander and task formalization but expands the question in a sequence as shown in Figure 2. We SFT Qwen2.5-32B, Qwen2.5-72B, and QwQ on both datasets. Other training settings remain the same. We denote method with the layer-wise structure as L, while the sequential

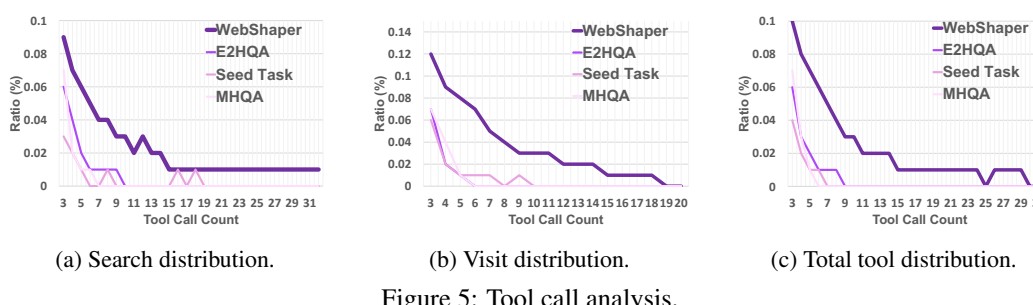

(a) Search distribution. (b) Visit distribution. (c) Total tool distribution.

Figure 5: Tool call analysis.

structure as S. The results as shown in Figure 3b. The layer-wise structure performs better than the Sequential structure in all base models. The results show that our method truly mitigates shortcomings such as Redundancy and Reasoning shortcuts. Our method improves the final performance via the controllable structures.

### 4.4.5 TOOL CALL ANALYSIS

We show the distribution tool call count of the agent to solve a question in different datasets. We illustrate the tool call counts larger than 3, which shows the complicated trajectories proportion. **Search Complexity (Figure 5a)** WebShaper exhibits a pronounced long-tail distribution. Pretty much tasks requiring over 3 search operations. This is 3-4x higher than E2HQA and MHQA, indicating superior handling of information-rich queries requiring iterative refinement. **Knowledge Navigation (Figure 5b)** The visit operation distribution shows WebShaper maintains a high ratio for trajectories exceeding 3 steps, while competing datasets sharply drop after 10 steps. This sustained capability reflects enhanced navigational intelligence in IS tasks. **Composite Reasoning (Figure 5c)** In total tool calls, WebShaper's doubles the count larger than 3. Notably, it sustains non-zero proportions up to 30 tool calls, demonstrating scalability for highly complex compositional reasoning. These findings underscore WebShaper's unique ability to manage intricate reasoning chains, with statistically significantly higher proportions of multi-hop reasoning trajectories across all modalities. The sustained performance in extended tool call sequences suggests superior architectural capacity for managing complex task decompositions compared to existing benchmarks.

### 4.5 COST ANALYSIS

Our overall synthesis pipeline is indeed more computationally demanding than traditional information-driven methods. On average, generating a single example requires roughly 20 LLM completions, 6 search calls, 6 visit calls, and about 7 minutes of end-to-end runtime. While approaches such as WebWalkerQA and MHQA incur lower cost by relying on only a few LLM completions, they typically yield simpler multi-hop questions. More advanced methods like E2HQA also rely on multiple LLM and tool calls and are therefore not substantially cheaper when targeting complex reasoning. Importantly, the additional compute is necessary to produce in-domain, high-fidelity GAIA and WebWalker-level data. As shown in Table 2, SFT on our synthesized data improves GAIA performance by 5–10 points across multiple backbones, consistently outperforming WebWalkerQA, E2HQA, and MHQA. These results underscore that the increased computational cost is well justified by the significant gains in data quality and downstream performance.

### 4.6 QUANTITATIVE EVALUATION OF QA FACTUAL ACCURACY

For the quantitative evaluation of QA factual accuracy, manual verification of large-scale synthesized QA pairs (thousands of instances) is impractical and resource-intensive. We thus adopt a proxy method leveraging the interpretability of our formalization: feeding the formalized structure (including intermediate reasoning steps) of each synthesized question to the QwQ-based solving agent, which provides explicit reasoning path guidance and makes the agent's answer accuracy a reliable reflection of the original QA pair's factual correctness. Experimental results show the agent achieves over **80%** accuracy on the synthesized dataset, confirming that our formalization-driven synthesis effectively mitigates hallucinations and inconsistencies while ensuring strong factual correctness.

### 4.7 Ethic Discussion

Our synthesis pipeline is specifically engineered to mitigate ethical concerns and data quality risks through three core safeguards. First, seed questions are derived from factual, rigorously curated Wikipedia entries, which effectively mitigates topic drift and minimizes the propagation of inherent biases. Second, the generated tasks are strictly centered on fact-driven reasoning, deliberately steering clear of subjective or sensitive domains, where biases are prone to emerge and amplify. Third, our multi-stage verification process systematically uphold factual consistency, while filtering out hallucinatory content and data contaminated by misinformation or bias.

## 5 Related Work

### 5.1 Information-Seeking Data Synthesis

Recent advances in information-seeking agents aim to integrate web interaction into LLMs' reasoning. While these works exhibit promising capabilities, they predominantly depend on limited or overly simplistic datasets (Yang et al., 2018; Joshi et al., 2017; Kwiatkowski et al., 2019). Concurrently, several recent benchmarks, such as GAIA (Mialon et al., 2023), BrowseComp (Wei et al., 2025), and BrowseComp-zh (Zhou et al., 2025), provide only test sets, which restricts their applicability for training agents. Early efforts, such as WebWalkerQA, explored simulating human-like web navigation to generate QA pairs by constructing linear information chains. CRAWLQA within WebDancer expands simple questions to more complex ones by aggregating external information, while SailorFog-QA within WebSailor leverages entity coreference networks to support fuzzy reasoning. These methods are predominantly information-driven, focusing on strategies for retrieving and connecting knowledge. In contrast, our approach is formalization-driven, emphasizing the structural representation and principled modeling of the QA process.

### 5.2 Formalization-based Data Synthesis

Recent work exploits formalization to synthesize training corpora for LLM theorem provers. DeepSeek-MathProver translates high-school and undergraduate competition problems into Lean4 statements, generates proofs with an LLM, and validates them in the Lean4 kernel (Xin et al., 2024). DeepSeek-MathProverV2 further decomposes proofs into subgoals and distills subgoal proofs into a lightweight model (Ren et al., 2025). Concurrently, Leang et al. (2025) synthesize "prover-as-judge" data via iterative natural-language↔formal-language alignment, replacing human feedback in RLHF and improving DPO outcomes. Goedel-Prover bootstraps a sequence of successively stronger provers on a dynamically expanding Lean4 corpus (Lin et al., 2025). A parallel line applies formalization to KBQA. LACT constructs arbitrary first-order logical queries via binary-tree decomposition, yielding an SFT dataset that is fine-tuned on an easy-to-hard curriculum (Xia et al., 2025). Departing from propositional or FOL formalisms, our work grounds data synthesis in set-theoretic IS.

## 6 Conclusion

This work presents a paradigm-shifting framework for synthesizing training data WebShaper for information-seeking (IS) agents through formalization-driven design. By establishing a set theory-based mathematical formalization of IS tasks, we address critical limitations in existing information-driven approaches that suffer from structural inconsistencies, task controllability, diversity, and coverage. The composition of proposed Knowledge Projections enables precise engineering of reasoning structures and complexity. Our agentic Expander module further ensures systematic expansion of formalized tasks with a layer-wise expansion paradigm, combining autonomous knowledge retrieval and rigorous validation to minimize redundancy and prevent reasoning shortcuts. Experimental results demonstrate that WebShaper not only achieves state-of-the-art performance on GAIA and WebWalkerQA benchmarks but also introduces controllability over task design, enabling deliberate engineering of cognitive challenges for IS agents. This formalization-driven paradigm shifts the focus from reactive information organization to proactive task specification, opening new avenues for advancing agent capabilities.

## Acknowledgements

This work is supported by National Natural Science Foundation of China (92470121, 62402016), National Key R&D Program of China (2024YFA1014003), Zhongguancun Academy (Grant No.s C20250204, C20250602), and High-performance Computing Platform of Peking University.

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

## A  DECLARATION ON THE USE OF LLMS

We affirm that the use of large language models in preparing this manuscript was strictly confined to language-related assistance, including sentence refinement and grammatical correction. All substantive content was independently authored by the authors and subsequently subjected to rigorous review and verification following any LLM-assisted edits. In conducting the experiments, LLMs were employed exclusively for legitimate academic research purposes, with no inappropriate applications. Detailed experimental settings are provided in Sec. 4 of this paper. Beyond the aforementioned language and experimental uses, no other reliance on LLMs was involved in this work.

## B  ILLUSTRATION OF FORMALIZATION-DRIVEN

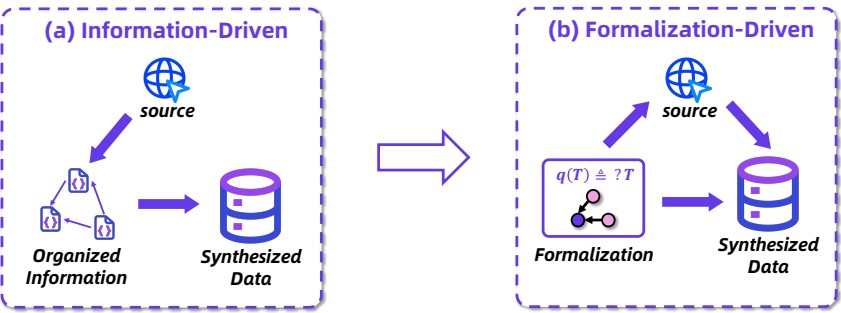

Figure 6: Data synthesis paradigm shift from information-driven to formalization-driven. "Source" stands for information sources such as the internet and databases. "Data" represents the synthesized QA data. (a) Previous methods retrieve and organize collected information in advance, then synthesize data according to the information structures. (b) Our method establishes the task formalization first, then collects information, and synthesizes QA data based on the formalization.

Compared with the traditional *information-driven* paradigm (Figure 6(a), which first collects raw information from various sources (e.g., the Internet and databases), organizes it into structured forms, and subsequently synthesizes QA data according to these information structures, our *formalization-driven* paradigm (Figure 6(b) begins by establishing a formalized representation of the target task (e.g., logical or symbolic specification). Guided by this formalization, we then acquire relevant information from sources and synthesize QA data directly in alignment with the established formal specification. This shift in paradigm emphasizes precise task modeling prior to information retrieval, enabling more controlled and consistent data generation.

Our data synthesis framework presents a foundational methodology for constructing training data for intelligent agents, featuring two key innovations: **task formalization** and **agent-driven synthesis**. By explicitly modeling tasks as structured, formal representations and leveraging proxy agents to synthesize data, this work provides a systematic approach to address the critical challenge of generating training data that transcends the complexity and unpredictability of naturally occurring human-centric environments. Below, we discuss the broader implications for agent research.

**Implications in Agent Training Data Synthesis**   Traditional approaches to training agents often rely on datasets derived from human-generated interactions, which are inherently limited in diversity, scalability, and controllability. We emphasize that effective agent training requires **explicit formalization of task structures**—a prerequisite for achieving precise control over data properties. By decoupling task definitions from data generation, the framework enables:

- *Targeted Complexity Management*: Tasks can be systematically parameterized to adjust difficulty, modality, or compositional structure, ensuring agents are exposed to controlled gradients of challenge. This contrasts with ad-hoc methods that risk overfitting to biases in natural data or failing to stress-test edge cases.
- *Quality Assurance*: Formal task models act as a "specification" for data synthesis, reducing noise and ensuring consistency. This is critical for applications where reliability and safety are paramount, such as autonomous systems or medical AI.

- *Scalable Data Generation*: Agent-driven synthesis eliminates the need for laborious manual annotation or heuristic-based pipelines by directly translating formal task representations into training instances. This reduces computational overhead while preserving fidelity to the task's intended design.

**Implications for AI Research and Development**   Our architecture provides insights for advancing AI systems:

- *Beyond Human-Level Complexity*: By formalizing tasks independent of human behavioral priors, the framework enables training data to exceed the implicit constraints of natural data. This opens pathways to train agents for domains requiring superhuman reasoning (e.g., advanced scientific modeling, combinatorial optimization).

- *Cross-Domain/Task Generalization*: Formal task representations abstract away domain-specific noise, allowing agents to learn invariant principles applicable across diverse contexts.

## C    PROOF OF PROPOSITION 1

Here we provide formal proof of Proposition 1.

*Proof.* Let $x$ be an element of $R(S_1) \cup R(S_2)$. By Equation 1, there exists either a $y_1 \in S_1$ such that $(y_1, x) \in R$ or $(x, y_1) \in R$, or a $y_2 \in S_2$ such that $(y_2, x) \in R$ or $(x, y_2) \in R$. Consequently, there exists a $y \in S_1 \cup S_2$, e.g., $y_1$ or $y_2$, such that $(y, x) \in R$ or $(x, y) \in R$. Thus, we have $x \in R(S_1 \cup S_2)$, and hence $R(S_1) \cup R(S_2) \subseteq R(S_1 \cup S_2)$.

Conversely, let $z$ be an element of $R(S_1 \cup S_2)$. Then there exists a $y \in S_1 \cup S_2$ such that $(y, z) \in R$ or $(z, y) \in R$. If $y \in S_1$, then $z \in R(S_1)$; if $y \in S_2$, then $z \in R(S_2)$. In either case, $z \in R(S_1) \cup R(S_2)$. Therefore, $R(S_1 \cup S_2) \subseteq R(S_1) \cup R(S_2)$.

Combining both directions, we conclude that:
$$R(S_1) \cup R(S_2) = R(S_1 \cup S_2).$$

Thus, we end proof of the Proposition. $\square$

## D    DATA STATISTICS

We analyze the domain distributions of our dataset. The domain distribution of our dataset demonstrates rather comprehensive coverage across multiple thematic areas, as visualized in Figure 7. Our construction of seed tasks leads to questions about various topics and entities. Our agentic expansion further strengthens these benefits. The dataset achieves significant diversity through its balanced representation of major domains such as `Sports`, `Politics`, and `Entertainment`.

This deliberate design ensures our dataset not only avoids over-reliance on any single domain but also maintains sufficient sample density across diverse topics. The empirical balance between breadth and depth enables robust training of a domain-agnostic information-seeking agent. Such characteristics position our dataset as particularly suitable for train multi-domain IS tasks and fostering interdisciplinary research.

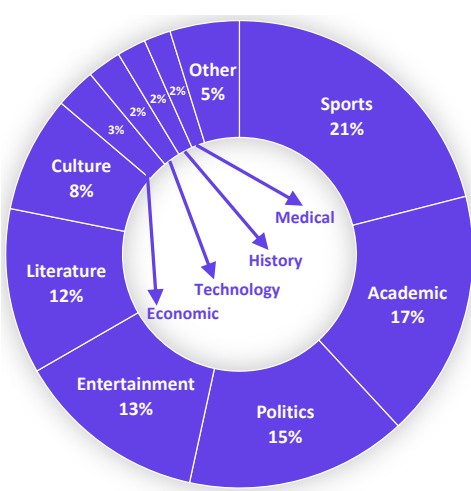

Figure 7: Domain distribution.

# E COMPARED DATASETS

We compare our synthesized dataset with several datasets:

- `WebWalkerQA` employs random walks over interlinked URLs to synthesize questions based on the visited webpages (Wu et al., 2025b). The dataset includes both single-source questions, generated from a single visited URL, and multi-source questions, which are constructed using information aggregated from multiple visited URLs.
- `E2HQA` is a dataset introduced by WebDancer (Wu et al., 2025a), where simple questions are systematically rewritten into more complex, challenging ones.
- `MHQA` is a composite dataset that integrates existing single-hop and multi-hop question-answering datasets. The majority of the questions are annotated by humans.

# F AGENT TRAINING

To train our information-seeking agent, similar to WebDancer (Wu et al., 2025a), we implement supervised fine-tuning (SFT) followed by reinforcement learning (RL).

In SFT, given a trajectory in a sequence of tokens $\mathcal{T} = (\tau_1, \alpha_1, o_1, ..., \tau_n, \alpha_n, o_n)$, we mask out loss from observation leading to loss:

$$L = -\frac{1}{\sum_{i=1}^{|\mathcal{T}|} \mathbb{I}[x_i \in o]} \sum_{i=1}^{|\mathcal{T}|} \mathbb{I}[x_i \in o] \cdot \log \pi_\theta(x_i \mid x_{<i}) \tag{8}$$

where $\pi_\theta$ is the model to be trained. Later in RL, we further optimize $\pi_\theta$ based on the GRPO algorithm (Shao et al., 2024). For a question-answer pair $(q, a)$, we sample rollouts $\{y_i\}_i^{|G|}$ and update the policy model by:

$$\mathcal{J}(\theta) = \mathbb{E}_{q \sim \mathcal{D}, \{y_i\}_{i=1}^G \sim \pi_{\theta_{old}}(\cdot|context)}$$

$$\left[ \frac{1}{\sum_{i=1}^G |y_i|} \sum_{i=1}^G \sum_{t=1}^{|y_i|} \min\left( r_{i,t}(\theta)\hat{A}_{i,t}, \ \text{clip}\left(r_{i,t}(\theta), 1 - \varepsilon_{low}, 1 + \varepsilon_{high}\right)\hat{A}_{i,t} \right) \right] \tag{9}$$

$$r_{i,j}(\theta) = \frac{\pi_\theta(o_i \mid q_i, o_{i,<t})}{\pi_{\theta_{old}}(o_i \mid q_i, o_{i,<t})}, \quad \hat{A}_{i,j} = \frac{R_i - \text{mean}(\{R_i\})}{\text{std}(\{R_i\})},$$

where $context$ includes all the model completions and tool responses. $\varepsilon$ is the clipping range of the importance sampling ratio $r_{i,t}(\theta)$. $\hat{A}_{i,t}$ is an estimator of the advantage of the $i$-th rollout at $t$-th step.

# G AGENT DETAILS

Following Wu et al. (2025a), WebShaper uses two tools, *search* and *visit*, which are regarded as fundamental to the information seeking process (Zhu et al., 2025):

- **Search** interfaces with the Google search engine to retrieve relevant documents given natural language queries. It supports multiple queries in parallel and returns the top-10 results for each query, where each result includes a title, a snippet, and the corresponding URL.
- **Visit** enables targeted extraction from specific web pages. Each page is paired with a designated visit goal. The full content of the page is first retrieved using Jina (Jina.ai, 2025), after which a summarization model (Qwen-2.5-72B in our implementation) extracts information relevant to the specified goal.

# H TRAINING DETAILS

## H.1 SFT

For SFT, we use a batch size of 32 and a learning rate of 5e-6, warmup plus cosine decay schedule. We also apply a weight decay of 0.1.

## H.2  RL

For RL training (Sheng et al., 2025), each group consists of 8 rollouts. The temperature is 1.0, $top_p = 1.0$, the batch size is 128, the mini batch size is 32, and the learning rate is 1e-6.

### H.2.1  CASE STUDY

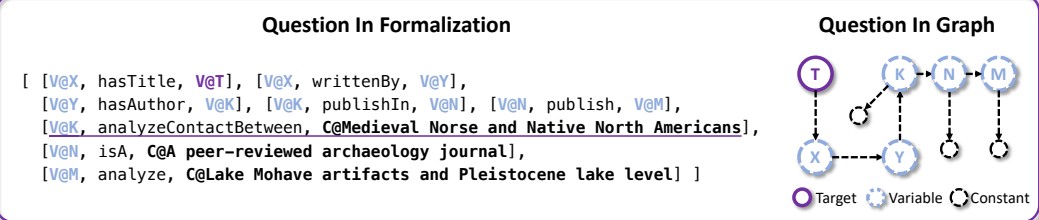

**Question In Natural Language**

**Question:** What is the title of the section, where the section is written by an author who also authored a scholarly article analyzing contact between Medieval Norse and Native North Americans published in a peer-reviewed archaeology journal, which additionally published another article that analyzes Lake Mohave artifacts and Pleistocene lake levels?
**Answer:** Thule Prehistory of Canada.

**Question In Formalization**

```
[ [V@X, hasTitle, V@T], [V@X, writtenBy, V@Y],
  [V@Y, hasAuthor, V@K], [V@K, publishIn, V@N], [V@N, publish, V@M],
  [V@K, analyzeContactBetween, C@Medieval Norse and Native North Americans],
  [V@N, isA, C@A peer-reviewed archaeology journal],
  [V@M, analyze, C@Lake Mohave artifacts and Pleistocene lake level] ]
```

**Question In Graph**

Target ⬤  Variable ⬡  Constant ⬡

Figure 8: Case studies of our synthesized data. We show a question in natural language, our formalization, and a graph respectively.

We present a representative case study in Figure 8. Compared with linear structure and sequential structure, our synthesized data has no problems of redundancy and reasoning shortcuts. The model should strictly seek information and reason alongside all the variables to find the answer. There are no constants directly connected to the target variable $T$ or variables close to it. Besides, there are no constants connected to other constants.

Moreover, $R$-Union effects well in our data. The underlined FP is a summarization of distributed web contents, leading to more difficulty in resolving the variables $K$, $N$, and $M$. Benefiting from the formalization, our data contains a variety of IS forms, which can fully stimulate the different IS capabilities of the model.

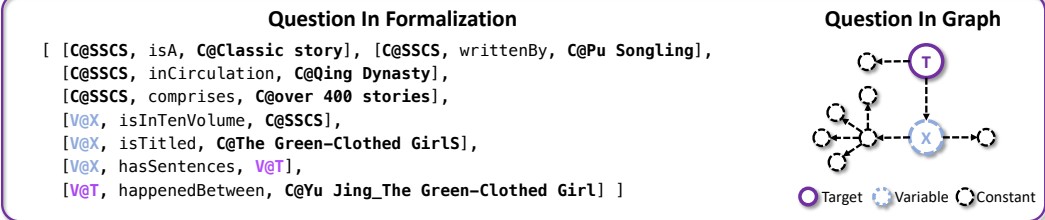

**Question In Natural Language**

**Question:** "Strange Stories from a Chinese Studio" is a collection of classical Chinese short stories written by the Qing Dynasty novelist Pu Songling. The earliest manuscript copies were already in circulation during the Kangxi reign of the Qing Dynasty, and the collection comprises over four hundred short stories in total. In Volume Ten of "Strange Stories from a Chinese Studio," there is a story titled "The Green-Clothed Girl." In this story, how many sentences did the scholar Yu Jing speak with her?

**Question In Formalization**

```
[ [C@SSCS, isA, C@Classic story], [C@SSCS, writtenBy, C@Pu Songling],
  [C@SSCS, inCirculation, C@Qing Dynasty],
  [C@SSCS, comprises, C@over 400 stories],
  [V@X, isInTenVolume, C@SSCS],
  [V@X, isTitled, C@The Green-Clothed GirlS],
  [V@X, hasSentences, V@T],
  [V@T, happenedBetween, C@Yu Jing_The Green-Clothed Girl] ]
```

**Question In Graph**

Target ⬤  Variable ⬡  Constant ⬡

Figure 9: Case comparison. "SSCS" stands for "Strange Stories from a Chinese Studio".

We compare a representative example shown by KIMI-Researcher (Kimi, 2025), illustrated in Figure 9. The case includes redundant information, such as multiple constants connected to "SSCS", which contribute little to answering the question. Additionally, a reasoning shortcut is observed that directly connects to the target variable. Despite the apparent complexity, the underlying reasoning structure is relatively simple, consisting of a single-hop reasoning path.

Table 3: **SFT Data Comparison** on GAIA benchmarks. The best results among all backbones are in **bolded**.

| Backbone | Dataset | GAIA | | | |
|---|---|---|---|---|---|
| | | Level 1 | Level 2 | Level 3 | Avg. |
| Qwen-2.5-32B | WebWalkerQA | 43.5 | 30.7 | 0.0 | 32.0 |
| | E2HQA | 56.4 | 36.5 | 0.0 | 39.8 |
| | MHQA | 43.5 | 36.5 | 8.3 | 35.9 |
| | **WebShaper** | 56.4 | 40.3 | 16.6 | **43.6** |
| Qwen-2.5-72B | WebWalkerQA | 53.8 | 36.5 | 0.0 | 38.8 |
| | E2HQA | 61.5 | 38.4 | 16.6 | 44.6 |
| | MHQA | 56.4 | 44.2 | 0.0 | 43.6 |
| | **WebShaper** | 56.4 | 48.0 | 0.0 | **45.6** |
| QwQ-32B | WebWalkerQA | 66.6 | 38.4 | 8.3 | 45.6 |
| | E2HQA | 58.9 | 42.3 | 16.6 | 45.6 |
| | MHQA | 51.2 | 44.2 | 0.0 | 41.7 |
| | **WebShaper** | 69.2 | 50.0 | 16.6 | **53.3** |

# I  DETAILED DATA COMPARISON RESULTS

As shown in Table 3, our proposed WebShaper method consistently achieves the highest average performance across different backbones on the GAIA benchmarks. In particular, WebShaper outperforms other datasets in most settings, with the best results highlighted in bold. For example, with the Qwen-2.5-32B backbone, WebShaper achieves an average score of 43.6, surpassing competing datasets by a significant margin. Similarly, for Qwen-2.5-72B and QwQ-32B backbones, WebShaper reaches 45.6 and 53.3 respectively, demonstrating strong generalization capabilities across model sizes and difficulty levels (Level 1, Level 2, Level 3). These results clearly highlight the robustness and superiority of our approach in handling diverse and challenging evaluation settings.

