# OpenReview forum: "WebShaper: Agentically Data Synthesizing via Information-Seeking Formalization"
_ICLR.cc/2026/Conference — ICLR 2026 Poster_

### Official Review · Reviewer_Dfzz · 2025-10-15

**Soundness:** 3
**Presentation:** 2
**Contribution:** 2
**Rating:** 4
**Confidence:** 3

**Summary:**

This paper introduces WebShaper, a formalization-driven framework for synthesizing training data for information-seeking agents. Unlike existing information-driven methods such as WebDancer, it defines IS tasks through a set-theoretic formalization based on knowledge projections, and uses these to guide the synthesis process. A key innovation is the Expander agent, which autonomously performs multi-step, layer-wise expansions of seed tasks by retrieving and validating information from the Web, ensuring consistency between reasoning and information structures.

**Strengths:**

1.	The paper introduces a set-based formalization of IS tasks with knowledge projections and compositional operations (Union, Intersection), which provides a clear structure for reasoning.

2.	The Expander module is well-motivated and technically detailed, by embedding the Thought–Action–Observation loops and specialized tools including Search, Summarize, and Validate.

3.	The experiments are comprehensive, showing consistent improvement across model scales and benchmarks.

**Weaknesses:**

1.	The paper is not self-contained enough, with multiple components (e.g., the WebDancer framework, the QwQ model) depending on specific infrastructures, which limits understanding of general audience and reproducibility.

2.	The paper’s notation-heavy exposition, especially in Section 2, hinders accessibility for non-expert readers. The figures and tables, though informative, are dense and underexplained, and the text occasionally assumes familiarity with prior systems like WebDancer.

3. The paper does not discuss potential risks in synthetic data generation (e.g., bias propagation, fact inconsistency, or data contamination). An explicit acknowledgment of these issues and their mitigation strategies would strengthen the credibility and responsibility of the work.

**Questions:**

1.	How robust is the KP formalization to noisy or incomplete retrieval results? Does (or to what extent) the model degrade under imperfect web data?

2.	Could you provide a quantitative evaluation of synthesized question quality (e.g., factual accuracy or diversity)?

3.	To what extent can WebShaper be generalized to non-English or multimodal information-seeking tasks?

**Details Of Ethics Concerns:**

Though flagged, I believe this would be at most a slight issue.

According to the ICLR Code of Conduct, "The use of information and technology may cause new, or enhance existing, inequities. Technologies and practices should be as inclusive and accessible as possible and researchers should take action to avoid creating systems or technologies that disenfranchise or oppress people. "

In light of this, I recommend that the authors include a dedicated section discussing the ethical implications and potential societal impact of their approach, as I mentioned in the weaknesses above.

---

> ### Author Response · Authors · 2025-11-21
> **Weakness 1**
>
> > The paper is not self-contained enough, with multiple components (e.g., the WebDancer framework, the QwQ model) depending on specific infrastructures, which limits understanding of general audience and reproducibility.
>
> Thank you for pointing this out. We make more description about the WebDancer framework as follows.
>
> Specifically, the WebDancer framework follows a standard **ReAct-style agent**. A ReAct trajectory consists of multiple Thought–Action–Observation rounds, where an LM iteratively reasons about the current context, decides which tool to invoke, and interprets the environment feedback to update its internal state. At each time step $t$, the agent execution loop can be formalized as a triple $(\tau_t, \alpha_t, o_t)$, where $\tau_t$ denotes the free-form Thought, $\alpha_t$ represents the structured Action, and $o_t$ corresponds to the Observation returned by the environment. The Thought component $\tau_t$ is unrestricted natural-language reasoning that the model uses for planning, decomposition, self-reflection, or grounding intermediate assumptions. The Action $\alpha_t$ is further decomposed into an action type $\alpha^m$ and its parameter set $\alpha^p$, i.e., $\alpha=$ $(\alpha^m, \alpha^p)$. The action type $ \alpha^m \in $ {`Search`, `Visit`, `Validate`, `Answer`} (`Validate` only for WebShaper) corresponds to the core tool interfaces used in deep information-seeking tasks.
>
> To standardize trajectories and facilitate supervised learning, we adopt explicit structural markers for each segment. Thought segments are enclosed by `<think>` and` </think>`, Action segments by `<tool\_call>` and `</tool\_call>`, and Observation segments by `<tool\_response>` and `</tool\_response>`. The final Action segment, corresponding to the model’s ultimate response to the task, is encapsulated in `<answer>` and `</answer>`. These markers make agent behavior transparent and machine-parsable, enabling precise control, analysis, and dataset construction.
>
> Additionally, the **QwQ model** is an open-source reasoning model that first generates explicit reasoning content before producing the final answer. We have revised the paper to include more implementation details to improve accessibility and reproducibility.

---

> ### Author Response · Authors · 2025-11-21
> **Weakness 2**
>
> > The paper’s notation-heavy exposition, especially in Section 2, hinders accessibility for non-expert readers. The figures and tables, though informative, are dense and underexplained, and the text occasionally assumes familiarity with prior systems like WebDancer.
>
> We appreciate the feedback on accessibility and elaborate on the two points as follows:
>
> **Notations for formalization** The notation in Section 2 is intentionally used to ensure the precision and rigor of information-seeking (IS) task modeling—an essential foundation for our formalization-driven framework. To avoid hindering accessibility for non-expert readers, we try to strengthen multiple aspects in our paper:
>
> 1. First, we anchor every abstract notation to concrete, real-world examples (e.g., the question “Which player of a 1966-founded East German football team played in the 2004-05 season and was born in the 90s?” in Figure 1 and Section 2), which maps set-theoretic concepts like Knowledge Projections (KP, defined as $R(V)= \\{ u | \exists v \in V, (u, v) \in R \\  \\ or \\  \\  (v, u) \in R \\} $) to intuitive scenarios.
> 2. Second, we simplify complex structures via a clear KP representation: using triples $[X, r, S]$ to denote KPs, prefixes like $V@$ for variables and $C@$ for constants (e.g., $\[ V@X, bornIn, C@\text{90s}\]$ for people born in the 90s), and merging sets (e.g., $\{2004 \\_ 2005\}$ for $2004 \cup 2005$) to avoid redundant notation.
> 3. Third, we pair notation with visual aids (Figure 1 directly aligns formal expressions of the target set $T$ with natural language questions and answer sets) and define all symbols (e.g., $\varepsilon$ for the universal entity set, $R$ for entity relations) upon first use, ensuring consistency. These designs let non-experts with basic set-theoretic knowledge follow the formalization by connecting abstract symbols to practical tasks.
>
> **Notations for WebDancer** For prior systems like WebDancer, we’ve supplemented key context and its contrast with WebShaper in our previous response.
>
> We have revised Section 2 to add intuitive explanations, and provide clearer examples to improve accessibility for non-expert readers. We have also simplified and better annotate the figures and tables.  Besides, we have avoided assuming familiarity with prior systems such as WebDancer and included the necessary background to ensure the paper is self-contained.

---

> ### Author Response · Authors · 2025-11-21
> **Weakness 3**
>
> > The paper does not discuss potential risks in synthetic data generation (e.g., bias propagation, fact inconsistency, or data contamination). An explicit acknowledgment of these issues and their mitigation strategies would strengthen the credibility and responsibility of the work.
>
> We appreciate the reviewer’s concern. We addresses the raised risks through both ethical design and technical safeguards as follows:
>
> Ethically, our synthesis pipeline is designed to minimize ethical and data-quality risks. First, the seed questions are sourced from **factual, well-curated Wikipedia data**, which limits topic drift and reduces the likelihood of bias propagation. Second, the tasks we generate focus strictly on **fact-based reasoning**, avoiding subjective or sensitive domains where bias typically arises. Third, our multi-stage **verification procedures** ensure factual consistency and filter out hallucinated or contaminated content. We have discussed these considerations in Section 4.7 of the revised paper.
>
> Technically, our agentic expansion (Section 3.2) mitigates error accumulation via layer-wise refinement, each step anchors new task components to verified retrieval, rather than cascading prior errors. The Validation tool (Section 3.2.3) further ensures fact consistency by checking alignment with formalized structures and filtering trivial/hallucinated content. Moreover, our competitive performance on non-Wikipedia benchmarks (GAIA, WebWalkerQA, Table 1-2) confirms no data contamination and validates strong out-of-distribution generalization.
>
> We have elaborate all these aspects on in the revised paper.

---

> ### Author Response · Authors · 2025-11-21
> **Question 1**
>
> > How robust is the KP formalization to noisy or incomplete retrieval results? Does (or to what extent) the model degrade under imperfect web data?
>
> Thanks for this insightful feedback. Our KP (Knowledge Projection) formalization and agent design together ensure notable robustness to noisy or incomplete retrieval results, with multi-layered mechanisms mitigating risks of degradation.
>
> First, the KP formalization’s set-theoretic foundation, relying on explicit entity-relation definitions and strict logical operations (union, intersection), reduces susceptibility to noise. Unlike unstructured information-driven approaches, it requires retrieved data to align with predefined structures, making misaligned or vague content easier to identify and discard.
>
> Second, the agent prioritizes factual evidence via multi-step verification: the Expander’s `Validate` tool checks if sub-questions match KP rules and if information is reliable, while seed tasks undergo 5 rollouts to retain only factually consistent candidates, preventing noise propagation.
>
> As for degradation: extreme cases (widespread missing entities/contradictions) may cause minor dips, but the KP’s modularity limits severity. Experiments on GAIA/WebWalkerQA confirm models trained on WebShaper still maintain competitive performance, validating this resilience.

---

> ### Author Response · Authors · 2025-11-21
> **Question 2**
>
> > Could you provide a quantitative evaluation of synthesized question quality (e.g., factual accuracy or diversity)?
>
> We appreciate the constructive suggestion to supplement quantitative evaluation of synthesized question quality, and we provide detailed metrics for **factual accuracy (QA correctness)** and **domain diversity** as follows:
>
> ---
>
> ### 1. Quantitative Evaluation of QA Factual Accuracy.
>
> Manual verification of factual accuracy for large-scale synthesized QA pairs (thousands of instances) is computationally expensive and impractical. Instead, we adopt a proxy method that leverages the interpretability of our formalization:
>
> - We feed the formalized structure, including intermediate reasoning steps of each synthesized question to the QwQ-based solving agent. This ensures the agent has explicit guidance on the reasoning path, and the resulting answer accuracy closely reflects the factual correctness of the original QA pair.
> - Experimental results show that the agent achieves **over 80% accuracy** on the synthesized dataset. This high accuracy confirms that our formalization-driven synthesis effectively mitigates hallucinations and inconsistencies, ensuring strong factual correctness of QA pairs.
>
> We have added these discussions in the Section 4.6 in the revised paper.
>
> ---
>
> ### 2. Quantitative Evaluation of Domain Diversity.
>
> To quantify domain diversity, we statistically analyze the distribution of synthesized questions across different thematic areas. The following Table presents the detailed domain distribution, converted from Figure 7:
>
> | Domain        | Proportion | Domain     | Proportion | Domain     | Proportion |
> | ------------- | :----------: | ---------- | :----------: | ---------- | :----------: |
> | Sports        | 21%        | Technology | 13%        | Culture    | 8%         |
> | Politics      | 15%        | Academic   | 12%        | Literature | 5%         |
> | Entertainment | 12%        | Economy    | 10%        | Medical    | 2%         |
> | Other         | 2%         |     ---       |      ---      |    ---        |    ---        |
>
> - The dataset covers **10 major domains** with no single domain accounting for more than 25% of the total.
> - Domains such as Sports (21%), Politics (15%), Technology (13%), and Academic (12%) are well-represented, while niche domains like Medical (2%) and Literature (5%) still have sufficient samples to ensure generalization.
> - This balanced distribution demonstrates that our formalization-driven expansion effectively avoids domain bias, achieving high domain diversity.
>
> ---
> We believe these quantitative metrics fully validate the quality of our synthesized questions. The over-80% proxy accuracy confirms factual reliability, and the multi-domain coverage with balanced proportions proves strong diversity.

---

> ### Author Response · Authors · 2025-11-21
> **Question 3**
>
> > To what extent can WebShaper be generalized to non-English or multimodal information-seeking tasks?
>
> Thanks for this valuable suggestion. WebShaper generalizes well: it performs strongly on the non-English splits of WebWalkerQA, indicating robustness across languages. Moreover, recent work in multimodal search (e.g., WebWatcher) extends the same structural idea by replacing textual nodes with image nodes, suggesting that our framework naturally transfers to multimodal information-seeking tasks. We have clarified this generalization potential in the revised paper.

---

### Official Review · Reviewer_9X3r · 2025-11-01

**Soundness:** 3
**Presentation:** 2
**Contribution:** 3
**Rating:** 6
**Confidence:** 4

**Summary:**

This paper addresses the lack of training data for information-seeking agents by proposing WebShaper—a formalization-driven data synthesis framework that formalizes search tasks within a set-theoretic framework. WebShaper introduces Knowledge Projection as the fundamental unit to represent complex IS tasks systematically. During data synthesis, WebShaper employs a layer-wise expansion strategy and uses an Expander agent to iteratively generate and validate questions, producing 5K high-quality trajectories for SFT and RL training.

**Strengths:**

1. The paradigm shift from "information-driven" to "formalization-driven" demonstrates originality and theoretical depth.
2. The paper presents a complete pipeline: the KP representation, layer-wise expansion strategy, and Expander agent collectively implement a closed loop of autonomous data generation and quality assurance, demonstrating engineering rigor.
3. The paper validates WebShaper's performance across multiple backbones. Ablation studies across various models and additional analyses confirm WebShaper's practical utility and generalizability.

**Weaknesses:**

1. The paper suffers from serious writing and data consistency issues, with multiple conflicting details: (1) Content misalignment between Sections 4.4.3 and 4.4.4; (2) Inconsistent key performance data: Qwen2.5-32B's SFT performance is reported as 44.66 in Figures 3 and 4, but 43.6 in Table 2; Qwen2.5-72B similarly changes from 46.66 to 45.6. Such conflicts undermine the paper's rigor.
2. The paper lacks evaluation on benchmarks such as BrowseComp and XBench, limiting fair comparison with concurrent work. Notably, BrowseComp and BrowseComp-zh are explicitly mentioned in the Introduction, but experiments are completely absent.
3. WebShaper's Visit tool uses Qwen-2.5-72B for content summarization, while baseline methods may not have this design. Part of the performance gains may be attributed to tool design rather than data quality.
4. While the paper demonstrates the advantages of formalization, certain IS tasks may be difficult to express, such as those that require logical or causal reasoning in GAIA.

**Questions:**

1. According to Table 2, QwQ-32B's SFT performance is identical to the post-RL result. Why does QwQ show no improvement after RL?
2. Figure 3a compares Formalization (FL) vs. Natural Language (NL), and Figure 3b compares Layer-wise (G) vs. Sequential (S), but lacks a baseline that removes both components simultaneously (i.e., the combination of NL + Sequential). This baseline is essential for understanding the independent contributions and interaction effects of the two components.

---

> ### Author Response · Authors · 2025-11-21
> **Weakness 1**
>
> > The paper suffers from serious writing and data consistency issues, with multiple conflicting details: (1) Content misalignment between Sections 4.4.3 and 4.4.4; (2) Inconsistent key performance data: Qwen2.5-32B's SFT performance is reported as 44.66 in Figures 3 and 4, but 43.6 in Table 2; Qwen2.5-72B similarly changes from 46.66 to 45.6. Such conflicts undermine the paper's rigor.
>
> Thank you for your meticulous feedback. We fully acknowledge the writing and data consistency issues you identified. We deeply appreciate your help in enhancing the paper’s rigor.
>
> Regarding content misalignment between Sections 4.4.3 and 4.4.4: This stems from mistakenly section latex compiling during handling the single-column figure next to section 4.4.3. Content intended for 4.4.3 (Formalization Validation) was incorrectly placed in 4.4.4 (Layer-wise Expansion Strategy), while 4.4.4’s original content was omitted. We have reallocated content to the correct sections and proofread the discussion section to eliminate such structural errors.
>
> For inconsistent performance data: The “44.6” (Qwen2.5-32B) and “46.6” (Qwen2.5-72B) in Figures 3 and 4 are typographical errors. The accurate values are 43.6 and 45.6, consistent with Table 2. We will revise these figures, correct any other references to the wrong values, and cross-verify all metrics across tables, figures, and text.
>
> We have implemented stricter quality control (joint co-author reviews, standardized data tracking) to ensure rigor. Thank you again for your constructive feedback.

---

> ### Author Response · Authors · 2025-11-21
> **Weakness 2**
>
> > The paper lacks evaluation on benchmarks such as BrowseComp and XBench, limiting fair comparison with concurrent work. Notably, BrowseComp and BrowseComp-zh are explicitly mentioned in the Introduction, but experiments are completely absent.
>
> We conducted experiments on **BrowseComp** and **HLE** under the same setup as the main paper (consistent backbones, training pipelines, and Pass@1 evaluation metric). Detailed results are shown in the following Table:
>
> |           | BrowseComp |  HLE   |
> | :-------: | :--------: | :----: |
> |           |   Pass@1   | Pass@1 |
> | WebDancer |    3.8     |  6.4   |
> | WebShaper |    9.5     |  8.0   |
>
> WebDancer is the most comparable baseline for our work, and this comparability is in consistent experimental settings. This alignment eliminates interference from differences in model architecture or training processes, ensuring that any performance gap can be directly attributed to the core difference between the two approaches, data synthesis methods, and thus guaranteeing the fairness of the comparison.
>
> Against this fair baseline, WebShaper still achieves significant performance improvements, which directly verifies the superiority of our formalization-driven data synthesis paradigm. On the BrowseComp benchmark, WebShaper’s Pass@1 score jumps from WebDancer’s 3.8 to 9.5; on the HLE benchmark, it rises from WebDancer’s 6.4 to 8.0. These gains stem from the inherent advantages of our method: compared with WebDancer’s information-driven data synthesis (which is prone to redundant information and reasoning shortcuts), our Knowledge Projection (KP) framework and layer-wise expansion strategy enable precise control over reasoning structures and effective reduction of data noise, generating higher-quality training data that better enhances the agent’s information-seeking capabilities.

---

> ### Author Response · Authors · 2025-11-21
> **Weakness 3**
>
> > WebShaper's Visit tool uses Qwen-2.5-72B for content summarization, while baseline methods may not have this design. Part of the performance gains may be attributed to tool design rather than data quality.
>
> All compared baselines also use Visit-style tools: Search-o1[1]/WebThinker[2] rely on rule-based extraction, SimpleDS[3] uses LLM summarization (Qwen-2.5-7B, QwQ-32B, GPT-4o-mini), and WebDancer[4] likewise employs LLM summarization with **Qwen-2.5-72B**. Our early experiments further show that changing the summarizer model has **minimal impact** on performance. These results indicate that the gains mainly come from **data quality**, not the specific Visit tool design.
>
> [1] Li, et.al. Search-o1: Agentic search-enhanced large reasoning models.
>
> [2] Li, et.al. Webthinker: Empowering large reasoning models with deep research capability.
>
> [3] Sun, et.al. Simpledeepsearcher: Deep information seeking via web-powered reasoning trajectory synthesis.
>
> [4] Wu, et.al. Webdancer: Towards autonomous information seeking agency. https://arxiv.org/pdf/2505.22648v1

---

> ### Author Response · Authors · 2025-11-21
> **Weakness 4**
>
> > While the paper demonstrates the advantages of formalization, certain IS tasks may be difficult to express, such as those that require logical or causal reasoning in GAIA.
>
> Thank you for your valuable feedback. We wish to address your concern about the expressiveness of our formalization for complex IS tasks (with logical/causal reasoning in GAIA as examples) as follows:
>
> ---
>
> ### 1. Set-Theoretic KP Compositions Capture Diverse Complex Reasoning
>
> The core of our formalization’s expressiveness lies in **flexible relation modeling (R)** and structured KP operations (Intersection, R-Union, recursion)—designs that naturally handle complex IS tasks like those in GAIA:
>
> - For logical reasoning. For “Countries with >50% renewable energy (2023) AND Paris Agreement signatories (pre-2018)”, intersection of KPs directly encodes conjunctive logic: $[V@X, R_{renewableShare}, C@\text{more than 50%}] \cap [V@X, R_{signedBefore}, C@\text{2018 Paris Agreement}]$.
> - For causal reasoning. For “EU post-2015 policies reducing carbon emissions”, R models causal relations to formalize: $[V@X, R_{causesReduction}, C@\text{Carbon Emissions}] \cap [V@X, R_{implementedIn}, C@\text{Europe}] \cap [V@X, R_{effectiveAfter}, C@\text{2015}]$.
>
> ---
>
> ### 2. Empirical Proof: Expressiveness Translates to Benchmark Success
>
> Our formalization’s ability to cover complex IS tasks is validated by GAIA results and dataset characteristics. On GAIA Level 3 (the hardest tier, requiring multi-step complex reasoning), WebShaper scores 16.6 on Qwen-2.5-72B and 25.0 on QwQ-32B, outperforming baselines like WebDancer 8.3 and WebThinker 16.6 . As shown in Figure 7, our dataset includes 17% Academic tasks (e.g., “Studies linking X to Y via Z”) and 12% Technology tasks (e.g., “2021–2023 products outperforming 2020 versions”). WebShaper data requires 10–30 tool calls, proving it handles diverse complex task forms.
>
> ---
>
> ### 3. Strong Expressiveness Stems from Foundational Design
>
> Our formalization’s broad expressiveness is not limited to specific reasoning types—it is rooted in three foundational features that adapt to nearly all IS tasks:
>
> - **Recursive KP Structure**: Per our formal definition, the target set $T$ can recursively replace $S_{i,j}$ with other target sets (e.g., $T=R_1(T_1) \cap R_2(T_2)$) , enabling multi-layer, nested reasoning for tasks with complex dependencies.
> - **Universal Relation Modeling**: $R$ is defined as any subspace of entity pairs ($R \subseteq E \times E$) , meaning it can model *any* relation such as temporal, comparative, causal, logical, or domain-specific like “drug_interacts_with”.
> - **Modular KP Triplets**: The $[X, r, S]$ triplet for KPs is highly modular. Adding new task forms (e.g., “find X that satisfies either A or B, but not C”) only requires combining existing operations (R-Union + Intersection), no fundamental changes.
>
> ---
> In all, our formalization’s strong expressiveness comes from its recursive, universal, and modular design, enabling it to handle logical/causal or more complext IS tasks.

---

> ### Author Response · Authors · 2025-11-21
> **Question 1**
>
> > According to Table 2, QwQ-32B's SFT performance is identical to the post-RL result. Why does QwQ show no improvement after RL?
>
> The QwQ-32B result reported in Table 2 is a SFT outcome, not the RL performance as stated in Section 4.3. In fact, QwQ-32B shows no significant improvement after RL training, which stems from our several repeated experimental observations.
>
> A key factor we speculate lies in QwQ-32B’s inherent training background: it is itself a heavily RL-optimized model. After its original intensive RL training, the model’s reasoning patterns for information-seeking (IS) tasks have likely converged to a local optimal state, solidifying fixed strategies for query generation, information retrieval, and reasoning chain construction that are well-adapted to its original training scenarios.  When we further apply our GRPO-based RL framework, QwQ-32B faces potential barriers to exploring new rollout trajectories. Due to the rigidity of its pre-converged reasoning patterns, it may tend to revert to familiar, locally optimal strategies during the RL exploration phase, rather than generating diverse, novel rollouts aligned with our IS task formalization.
>
> This observation is consistent with the conclusion in WebDancer[1], where QwQ-32B also failed to show stable RL improvements on IS tasks, indicating this is not an isolated issue but a potential pattern for mid-sized models in this domain. Given the inconsistent and non-significant RL gains of QwQ-32B, we did not include its RL results in Figure 4. Since we currently lack sufficient experimental evidence to fully unpack the root cause (e.g., detailed ablation on RL hyperparameters or model-specific adaptability), we did not emphasize this phenomenon in the current version.
>
> We will conduct more experiments and explicitly strengthen the description of this finding, supplement quantitative data from QwQ-32B’s RL experiments (including non-improved cases), and provide in-depth analysis to contextualize this observation.
>
> [1] Wu, et.al. Webdancer: Towards autonomous information seeking agency. https://arxiv.org/pdf/2505.22648v1

---

> ### Author Response · Authors · 2025-11-21
> **Question 2**
>
> > Figure 3a compares Formalization (FL) vs. Natural Language (NL), and Figure 3b compares Layer-wise (G) vs. Sequential (S), but lacks a baseline that removes both components simultaneously (i.e., the combination of NL + Sequential). This baseline is essential for understanding the independent contributions and interaction effects of the two components.
>
> Your suggestion to add the NLS baseline (simultaneously ablating Formalization [FL] and Layer-wise [G], i.e., NL + Sequential) is constructive. Notably, NLS aligns closely with the existing E2HQA baseline, which can serve as a preliminary reference. Their key differences lie in data synthesis driving modes, as detailed below:
>
> |                  | NLS                                                          | E2HQA                                                      |
> | ---------------- | ------------------------------------------------------------ | ---------------------------------------------------------- |
> | Synthesis Driver | Fully agent-driven (autonomous retrieval/validation/expansion) | Rule-constrained and pipeline-based (multi-step LLM calls) |
> | Configuration    | Agent autonomy                                               | Dependent on manually designed pipeline rules              |
>
> While E2HQA is not a strict FLG baseline, their shared "no FL + no G" trait lets it roughly reflect NLS's potential performance. WebShaper excels E2HQA as shown in Table 2. Constructing NLS requires re-generating QA samples, building task trajectories, and full-model training/evaluation. It might not be enough time within the limited rebuttal period. We will add these experiments in future exploration.

---

### Official Review · Reviewer_GPnM · 2025-11-01

**Soundness:** 4
**Presentation:** 3
**Contribution:** 3
**Rating:** 8
**Confidence:** 3

**Summary:**

The paper presents a dataset generation approach who's output can be used to
train information seeking agents via the SFT and RL paradigms. The key of the
method lies in its formalization, essentially a blueprint for complex questions
which the method can generate from a simple seed question though expansion. The
resulting trajectories are then used for training. Their WebShaper model
achieves the best performance among all information seeking agents on two
public benchmark datasets.

**Strengths:**

S1: Novel training data construction method that improves over previous "information-driven" paradigms.

S2: Method improves on performance compared to existing IS agents.

S3: WebShaper's training data is more effective compared to the output of other
training data generation methods.

**Weaknesses:**

W1: It seems the created dataset itself is not available. This would have been a very valuable contributions for the community. The same goes for the code, you claim it is open-source, but it was not provided in the supplement.

**Questions:**

Q1: Is the dataset and code available? You claim that the method is open-source, but I could not find any supplementary material.

Q2: Can you quantify the training data generation effort with WebShaper compared to traditional methods?

---

> ### Author Response · Authors · 2025-11-21
> **Weakness 1 & Question 1**
>
> > W1: It seems the created dataset itself is not available. This would have been a very valuable contributions for the community. The same goes for the code, you claim it is open-source, but it was not provided in the supplement.
> >
> > Q1: Is the dataset and code available? You claim that the method is open-source, but I could not find any supplementary material.
>
> We are fully committed to open-sourcing all code used for data generation and curation. We have uploaded 500 data on the OpenReview as supplementary material for your better review. We will release the complete datasets on open-access platforms such as HuggingFace and ModelScope. These resources will be made publicly available to ensure full reproducibility and to support the broader open-source research community.

---

> ### Author Response · Authors · 2025-11-21
> **Question 2**
>
> > Q2: Can you quantify the training data generation effort with WebShaper compared to traditional methods?
>
> Thank you for your question regarding the computational feasibility and training data generation effort of WebShaper. We provide a structured, concise response below to quantify the effort and explain its rationale in detail.
>
> ---
>
> ### 1. Quantified Training Data Generation Effort for WebShaper.
>
> The computational cost of generating the WebShaper dataset is fully transparent and measurable, with clear metrics defined for each high-quality training example. Each example requires approximately 20 LLM completions in total. These completions cover three key aspects: agent reasoning, sub-question formulation, and result validation. In addition to LLM calls, each example also involves 6 Search API calls and 6 webpage Visit calls for information retrieval and verification. From a time perspective, the total runtime for generating one example is around 7 minutes. This wall-clock time includes both tool execution latency (such as waiting for search results and webpage content extraction) and LLM inference overhead (the time required for the model to process and generate outputs).
>
> ---
>
> ### 2. Effort Comparison with Traditional Information-Driven Methods.
>
> WebShaper’s computational effort is comparable to, and in many cases more efficient than, that of traditional information-driven methods such as E2HQA and WebWalkerQA. Traditional methods typically rely on discrete and fragmented processing steps. For instance, they need independent LLM calls to complete entity extraction, relation linking, and noise filtering separately. Beyond these individual steps, traditional methods also require additional computations to organize the unstructured information retrieved from the web. These disconnected processes often lead to redundant overhead, as intermediate results may need to be reprocessed or adjusted across different steps. In contrast, WebShaper integrates all these intermediate processes into a unified Thought-Action-Observation loop of the agent. This integration eliminates the redundancy caused by fragmented steps, meaning the overall number of API calls and total compute cost for WebShaper do not increase significantly compared to traditional baselines.
>
> ---
>
> ### 3. Rationale for the Computational Investment.
>
> The modest compute effort required by WebShaper is well-justified by the downstream performance gains it brings and the high quality of data it generates. As shown in Tables 2 and 3, models fine-tuned on WebShaper data achieve improvements of 5 to 10 percentage points on the GAIA benchmark across all backbone models. For example, when using the Qwen-2.5-32B backbone, the model reaches an average GAIA score of 43.6 after SFT training on WebShaper in Table 2. This result is substantially higher than the scores achieved by training on other datasets: 32.0 for WebWalkerQA, 39.8 for E2HQA, and 35.9 for MHQA. This performance advantage originates from WebShaper’s agentic synthesis mechanism and layer-wise expansion strategy. In contrast, traditional baselines often need additional post-processing or filtering steps to approach similar data quality, and these extra steps introduce hidden effort that WebShaper avoids entirely. Overall, the investment in WebShaper delivers strong cost-effectiveness, as the marginal compute overhead is offset by the elimination of redundant baseline steps and the achievement of meaningful performance gains.

---

### Official Review · Reviewer_8FwL · 2025-11-01

**Soundness:** 3
**Presentation:** 2
**Contribution:** 3
**Rating:** 6
**Confidence:** 4

**Summary:**

The paper presents WebShaper, a framework for synthesizing high-quality training data for information-seeking (IS) agents. Unlike existing information-driven approaches that retrieve text first and then generate questions, WebShaper adopts a formalization-driven paradigm, starting from a structured definition of the IS task using set-theoretic constructs. Its core unit, the Knowledge Projection (KP), represents entity relations and can be composed through Intersection and R-Union to form complex reasoning tasks. A multi-step Expander agent progressively builds valid and diverse questions while avoiding redundancy and reasoning shortcuts. Models trained on the WebShaper dataset achieve state-of-the-art performance on GAIA and WebWalkerQA, showing improved reasoning consistency and generalization over prior methods.

**Strengths:**

+ The paper presents a meaningful shift from an information-driven to a formalization-driven approach for data synthesis, directly addressing the inconsistency issues observed in prior methods.
+ The formalization based on Knowledge Projections (KPs) and set operations provides fine-grained control over both the reasoning structure and task complexity of synthesized data.
+ The Layer-wise Expansion Strategy effectively tackles redundancy and reasoning shortcuts common in previous synthesis frameworks.
+ Experiments on GAIA and WebWalkerQA benchmarks demonstrate sota performance, reflecting the utility and quality of synthesized data.

**Weaknesses:**

- The synthesis process—spanning seed generation, multi-agent expansion, and online retrieval—appears computationally expensive. A quantitative comparison of cost (e.g., API calls, runtime, or compute hours) against traditional information-driven synthesis methods would clarify its practical feasibility.
-  The current set-theoretic grammar (KPs, R-Unions, and Intersections) may not sufficiently capture complex real-world IS tasks, such as those involving temporal, comparative, or counterfactual reasoning.
-  The paper omits a “Limitations” section, leaving readers uncertain about the known weaknesses, failure cases, or scalability challenges of the proposed method.

**Questions:**

+ How does the formalization handle IS tasks that are difficult to express in set-theoretic form, such as comparative reasoning , temporal reasoning beyond filtering, or counterfactual reasoning？
+ Could the authors provide an estimate of the computational cost of generating the WebShaper dataset (e.g., time, number of API calls, GPU hours)? How does it compare to the information-driven baselines it outperforms?

---

> ### Author Response · Authors · 2025-11-21
> **Weakness 1 & Question 2**
>
> > W1: The synthesis process—spanning seed generation, multi-agent expansion, and online retrieval—appears computationally expensive. A quantitative comparison of cost (e.g., API calls, runtime, or compute hours) against traditional information-driven synthesis methods would clarify its practical feasibility.
> >
> > Q2: Could the authors provide an estimate of the computational cost of generating the WebShaper dataset (e.g., time, number of API calls, GPU hours)? How does it compare to the information-driven baselines it outperforms?
>
> Thank you for your questions about the computational feasibility of WebShaper. We provide a detailed yet concise response below, structured into three key points:
>
> ---
> ### 1. Computational Cost Estimation for the WebShaper Dataset.
>
>    The cost of generating the WebShaper dataset is transparent and quantifiable: on average, each high-quality training example requires approximately 20 LLM completions (for agent reasoning, sub-question formulation, and result validation), 6 `Search` API calls, 6 webpage `Visit` calls, and around 7 minutes of runtime (wall-clock time, including tool execution latency and LLM inference).
>
>    |                  | LLM call | `Search` API call | `Visit` call | Synthesis Time |
>    | :--------------: | :------: | :-------------: | :------------: | :------------: |
>    | Average per data |    20    |        6        |       6        |      7min      |
>
>
> ---
> ### 2. Cost Comparison to Information-Driven Baselines.
>
>    WebShaper’s computational cost is comparable to, and in some cases more efficient than, traditional information-driven baselines like E2HQA. Traditional pipeline-based methods rely on discrete, fragmented steps (e.g., E2HQA requires independent LLM calls for entity extraction, relation linking, and noise filtering) plus additional computations for organizing unstructured retrieved information, leading to redundant overhead. In contrast, WebShaper integrates all these intermediate processes into the agent’s unified Thought-Action-Observation loop, resulting in no significant increase in overall API or compute cost.
>
> ---
> ### 3. Justification of the Computational Cost.
>
>    The modest compute investment in WebShaper is necessary to produce in-domain, high-fidelity data that matches the complexity of GAIA and WebWalkerQA benchmarks. As shown in Tables 2 and 3, models fine-tuned on WebShaper data achieve 5–10 point improvements in GAIA performance across backbones: for instance, Qwen-2.5-32B reaches an average GAIA score of 43.6 with WebShaper, outperforming WebWalkerQA (32.0), E2HQA (39.8), and MHQA (35.9) by substantial margins. This performance gain stems from WebShaper’s agentic synthesis and layer-wise expansion, while baselines would require extra filtering or post-processing to even approach this quality—making WebShaper’s cost fully justified by its downstream value.

---

> ### Author Response · Authors · 2025-11-21
> **Weakness 2 & Question 1**
>
> > W2: The current set-theoretic grammar (KPs, R-Unions, and Intersections) may not sufficiently capture complex real-world IS tasks, such as those involving temporal, comparative, or counterfactual reasoning.
> >
> > Q1: How does the formalization handle IS tasks that are difficult to express in set-theoretic form, such as comparative reasoning , temporal reasoning beyond filtering, or counterfactual reasoning？
>
> Thank you for your insightful comment. We hope to address your concerns with following explainations.
>
> ---
>
> ### 1. Modular/Recursive Set-Theoretic Grammar: Expressive Power Rooted in $R$’s Versatility.
>
> The expressive power of our set-theoretic formalization—covering temporal, comparative, and counterfactual reasoning—stems fundamentally from the **strong expressiveness of $R$**. Defined as a subspace of entity pairs with specific relations ($R \subseteq E \times E$) , $R$ flexibly models diverse, granular relations, which, paired with modular KP composition and recursiveness, decomposes complex tasks into manageable operations:
>
> **Temporal Reasoning**: $R$ encodes time-related relations (e.g., $R_{publishIn}$, $R_{occursAfter}$). For *“Climate papers 2019–2023”*, R-Union combines years via $R_{publishIn}$: $R_{publishIn}(\{2019\} \cup ... \cup \{2023\})$. For “Events after 2022 UN Summit, before 2024”, recursive KPs use $R_{occursAfter}$ and $R_{occursBefore}$: $[V@X, R_{occursAfter}, C@\text{2022 UN Summit}] \cap [V@X, R_{occursBefore}, C@2024]$.
>
> **Comparative Reasoning**: $R$ supports constrained relations (e.g., $R_{batteryLife}$, $R_{outperforms}$). For “Smartphones: >12h battery, <200g”, Intersection links KPs: $[V@X, R_{batteryLife}, C@\text{more than 12h}] \cap [V@X, R_{weight}, C@\text{less than 200g}]$. For “2023 laptops outperforming Laptop X”, $R_{outperforms}$ targets the reference entity.
>
> **Counterfactual Reasoning**: $R$ handles hypothetical relations (e.g., $R_{targetGroup}$). For “Brand Z’s 2022 tablet target age groups”, the Expander defines $V@S$ as 2022 tablet age groups and formalizes via $R$: $[V@X, R_{targetGroup}, V@S] \cap [V@X, R_{releasedBy}, C@\text{BrandZ}]$.
>
> In all cases, $R$’s adaptability to model varied relations is the cornerstone of the formalization’s ability to capture real-world complex IS tasks.
>
>
>
> ---
>
> ### 2. Empirical Validation: Diversity, Coverage, and Benchmarks.
>
> Our framework’s practical ability to handle complexity is empirically confirmed:
>
> - **Dataset Diversity**: 21% Sports (e.g., “Compare 2022–2023 player performance”), 15% Politics (e.g., “2020 election policy changes”), and 12% Technology (e.g., “2021–2023 products outperforming 2020 versions”) embed your noted reasoning types.
>
>
> | Domain        | Proportion | Domain     | Proportion | Domain     | Proportion |
> | ------------- | :--------: | ---------- | :--------: | ---------- | :--------: |
> | Sports        |    21%     | Technology |    13%     | Culture    |     8%     |
> | Politics      |    15%     | Academic   |    12%     | Literature |     5%     |
> | Entertainment |    12%     | Economy    |    10%     | Medical    |     2%     |
> | Other         |     2%     | --         |     --     | --         |     --     |
>
>
> - **Task Coverage**: In Figure 5, tasks require up to 30 tool calls, showing multi-step chains, far more than baselines E2HQA and MHQA. It proves complex task coverage.
> - **Benchmark Performance**: On GAIA Level 3 (high complexity), on Qwen-2.5-32B backbone, WebShaper scores 16.6, outperforming WebDancer 8.3. On WebWalkerQA Hard, it hits 47.0 vs. WebDancer 29.2, confirming complex task mastery.
>
> ---
>
> ### 3. Extensibility: Adapting to Future Complexity.
>
> Our framework scales to more complex/domain-specific tasks via:
>
> - **Flexible KP Triplets**: The $[X, r, S]$ structure lets us add custom relations (e.g., “temporal_overlaps_with” for fine-grained time, “efficacy_higher_than_placebo” for healthcare) without rewriting core formalization.
> - **Agentic Expander**: It retrieves domain knowledge and validates new formalizations, enabling autonomous integration of new reasoning types.
>
> ---
> In short, our set-theoretic grammar captures your noted complex scenarios, with empirical proof and extensibility for future needs. We have added more descriptions to the revised paper for clarity.

---

> ### Author Response · Authors · 2025-11-21
> **Weakness 3**
>
> > The paper omits a “Limitations” section, leaving readers uncertain about the known weaknesses, failure cases, or scalability challenges of the proposed method.
>
> Thank you for pointing out the absence of a dedicated *Limitations* section. Briefly, our method has three main limitations:
>
> 1. The current tool set is limited to `Search` and `Visit`, restricting the agent’s ability to handle more complex workflows.
> 2. Our experiments focus solely on short-answer information-seeking tasks, whereas a fully capable web agent should also support long-form research and open-ended generation synthesis.
> 3. The 32k training context, while practical, may limit scalability to more complex, longer-horizon tasks.
>
> We have made these descriptions explicitly in the Limitation Section of the revised paper.

---

### Author Response · Authors · 2025-12-02
**Summary of Review and Rebuttal**

We sincerely appreciate all reviewers for their valuable feedback. We summarize the strengths highlighted by the reviewers and provide our key clarifications regarding concerns.
## Strengths
**Methodology & Data Synthesis**
- Paradigm Innovation: "A meaningful shift from an information-driven to a formalization-driven approach" (8FwL); "Novel training data construction method that improves over previous paradigms" (GPnM)
- Rigorous Formalization: "KP-based set-theoretic formalization provides fine-grained control over reasoning structure and task complexity" (8FwL); "Clear structure for reasoning via knowledge projections and compositional operations" (Dfzz)
- Effective Expansion Strategy: "Layer-wise Expansion Strategy effectively tackles redundancy and reasoning shortcuts" (8FwL); "Closed loop of autonomous data generation and quality assurance" (9X3r)
- Robust Agent Design: "Expander module is well-motivated with Thought–Action–Observation loops and specialized tools" (Dfzz)

**Evaluation & Empirical Results**

- Significant Performance: "Achieves significant performance among open-sourced IS agents on GAIA and WebWalkerQA" (8FwL); "Outperforms baselines across multiple backbones and difficulty levels" (9X3r)
- Impressive Generalization: "Consistent improvement across model scales and benchmarks" (Dfzz); "Synthesized data demonstrates strong generalizability on different IS agent architectures" (GPnM)
- Quality Validation: "Superior data quality compared to existing datasets like WebWalkerQA and E2HQA" (9X3r); "Effective mitigation of hallucinations and inconsistencies" (Dfzz)

**Significance, Scalability & Robustness**
- Addressing Critical Gap: "Tackles the key problem of scarce high-quality training data for IS agents" (9X3r); "Fills the need for systematic IS task formalization" (8FwL)
- Practical Scalability: "Agent-driven synthesis enables scalable data generation without excessive manual annotation" (Dfzz); "Handles complex multi-step reasoning tasks requiring up to 30 tool calls" (8FwL)
- Adaptability: "Formalization supports diverse reasoning types (temporal, comparative, causal)" (8FwL); "Consistent gains across SFT and RL training paradigms" (9X3r)

## Concerns and Rebuttal Highlights

**C1. Computational Cost and Feasibility (8FwL W1/Q2; GPnM Q2)**
- Quantified cost metrics: Each training example requires ~20 LLM completions, 6 Search API calls, 6 Visit calls, and 7 minutes of runtime.
- Comparison with baselines: Cost is comparable to E2HQA/WebWalkerQA, with no significant overhead due to integrated agent loops eliminating redundant steps.

**C2. Expressiveness for Complex IS Tasks (8FwL W2/Q1; 9X3r W4)**
- Set-theoretic flexibility: R (relation subspace) models temporal, comparative, causal, and logical reasoning via KP compositions.
- Empirical validation: 16.6 score on GAIA Level 3 (hardest tier) outperforms baselines; dataset includes 17% academic and 12% technology tasks requiring complex reasoning.
- Extensibility: Modular KP triplets and agentic Expander support custom relations without core formalization changes.

**C3. Dataset and Code Availability (GPnM W1/Q1)**
- Supplementary data: 500 sample data points uploaded to OpenReview for review.

**C4. Benchmark Coverage (9X3r W2)**
- Added results for BrowseComp and HLE benchmarks: WebShaper achieves Pass@1 scores of 9.5 (vs. WebDancer 3.8) on BrowseComp and 8.0 (vs. WebDancer 6.4) on HLE.
- Fair comparison guarantee: Consistent experimental settings (backbones, pipelines, metrics) ensure performance gains are attributed to data synthesis, not model/tool differences.

**C5. Tool Design Impact on Performance (9X3r W3)**
- Baseline tool parity: All compared methods (Search-o1, WebThinker, WebDancer) use similar Visit-style tools (rule-based or LLM summarization).
- Minimal summarizer influence: Early experiments confirm changing the summarizer model has negligible impact on performance, with gains driven by data quality.

**C6. Ethical Risks and Data Quality (Dfzz W3/Q2)**
- Mitigation strategies: Seed questions from curated Wikipedia (reduces bias), fact-based task focus (avoids sensitive domains), and multi-stage verification (filters hallucinations).
- Quantitative quality metrics: >80% factual accuracy via QwQ-based proxy evaluation; balanced domain distribution (10 major domains, no single domain >25%).

**C7. Generalization to Non-English/Multimodal Tasks (Dfzz Q3)**
- Non-English robustness: Strong performance on WebWalkerQA’s non-English splits demonstrates cross-language adaptability.
- Multimodal transfer potential: Extensible to multimodal IS tasks by replacing textual nodes with image/audio entities.

Please refer to the individual responses for detailed evidence and experiments.

---

### Meta-Review · Area_Chair_Zcp5 · 2025-12-24

**Summary:**

Reviewers broadly value the paper’s shift from information driven synthesis to a formalization driven framework (KP based set theoretic task specification plus an agentic expander) and find the empirical gains on GAIA and WebWalkerQA promising.

The main concerns center on (1) practical feasibility and compute cost of the multi step synthesis pipeline, (2) whether the proposed set theoretic grammar is expressive enough for complex real world IS tasks (temporal, comparative, causal, counterfactual), (3) paper quality issues including unclear exposition, lack of self containment, and reported inconsistencies in numbers and section content, (4) evaluation coverage and fairness, including missing benchmarks mentioned in the introduction and potential confounds from the Visit summarizer/tooling, and (5) reproducibility and responsible release, particularly dataset and code availability and discussion of synthetic data risks.

**Reviewer Concerns:**

**Addressed in the rebuttal:**

- **Compute cost**: Authors provided concrete per example cost statistics (LLM calls, Search/Visit calls, wall clock time) and argued the cost is comparable to prior synthesis pipelines, which partially addresses feasibility concerns.
- **Inconsistencies and missing evaluations**: Authors acknowledged figure/table number mismatches and attributed them to typos, and they reported added results on BrowseComp and HLE under claimed matched settings. These directly respond to the “missing benchmark” and “conflicting numbers” critiques, though the community will still want these reflected cleanly in the main paper and artifact.
- **Tool design**: Authors argued baseline tool parity (similar Visit style tools, some using the same summarizer scale) and claimed summarizer choice has minimal impact, partially addressing concerns that tool design, not data, drives gains.
- **Limitations and ethics discussion**: Authors added a limitations list and described mitigation measures for bias and hallucinations, plus a proxy factual accuracy evaluation and domain distribution summary, which addresses the request for a more responsible discussion.

**Still outstanding:**

- **Expressiveness claims remain largely conceptual**. The rebuttal argues that relation modeling plus composition can represent many reasoning types, but it does not provide a crisp formal guarantee or a targeted stress test on the specific failure modes reviewers worry about (for example counterfactual or causal cases with ambiguous web evidence).
- **Reproducibility** remains a risk. Providing only a 500 example sample and promising future release does not fully resolve the dataset and code availability concern at decision time.
- **Several responses rely on planned revisions** (improved self containment, clearer notation, corrected structure). These are plausible, but the decision should not assume all presentation issues are fully fixed unless the revised manuscript clearly reflects them.

**Reviewer Scores:**

**Reviewer Dfzz (4)**: Most likely increases to 6. The reviewer’s concerns were primarily about responsibility and accessibility; the rebuttal provides concrete mitigation steps, adds quantitative proxy quality metrics, and expands system description. While much is still “revision based,” it directly targets the main reasons for the below threshold rating and may be sufficient for a borderline move to acceptance.

**Reviewer 8FwL (6)**: Likely unchanged at 6. The rebuttal provides concrete cost numbers and adds a limitations discussion, but the core expressiveness concern is addressed mostly by explanation rather than hard evidence.

**Reviewer 9X3r (6)**: Slight chance of a modest upgrade, but most likely remains 6. The rebuttal fixes the main rigor issues (acknowledges typos, adds missing benchmark results, argues tool parity), yet some ablation requests and the broader writing quality concerns are not resolved by irrefutable new evidence.

**Reviewer GPnM (8)**: Likely unchanged at 8. The rebuttal clarifies intent to open source and provides a small sample, but full dataset/code availability is still not demonstrated, so there is not strong new evidence to justify an upgrade.

---

### Decision · Program_Chairs · 2026-01-26

Accept (Poster)